# Conditioning sharpens the spatial representation of rewarded stimuli in mouse primary visual cortex

**Pieter M Goltstein[1,2†]\*, Guido T Meijer[1,2], Cyriel MA Pennartz[1,2]\***

[1]Center for Neuroscience, Swammerdam Institute for Life Sciences, University of Amsterdam, Amsterdam, Netherlands; [2]Research Priority Program Brain and Cognition, University of Amsterdam, Amsterdam, Netherlands

**Abstract** Reward is often employed as reinforcement in behavioral paradigms but it is unclear how the visuospatial aspect of a stimulus-reward association affects the cortical representation of visual space. Using a head-fixed paradigm, we conditioned mice to associate the same visual pattern in adjacent retinotopic regions with availability and absence of reward. Time-lapse intrinsic optical signal imaging under anesthesia showed that conditioning increased the spatial separation of mesoscale cortical representations of reward predicting- and non-reward predicting stimuli. Subsequent in vivo two-photon calcium imaging revealed that this improved separation correlated with enhanced population coding for retinotopic location, specifically for the trained orientation and spatially confined to the V1 region where rewarded and non-rewarded stimulus representations bordered. These results are corroborated by conditioning-induced differences in the correlation structure of population activity. Thus, the cortical representation of visual space is sharpened as consequence of associative stimulus-reward learning while the overall retinotopic map remains unaltered.
DOI: https://doi.org/10.7554/eLife.37683.001

**\*For correspondence:**
goltstein@neuro.mpg.de (PMG);
C.M.A.Pennartz@uva.nl (CMAP)

**Present address:** †Max Planck Institute of Neurobiology, Martinsried, Germany

**Competing interests:** The authors declare that no competing interests exist.

## Introduction

Involvement of sensory processing in perceptual and stimulus-outcome learning can alter stimulus selectivity of neurons in the sensory cortex (*Bao et al., 2001*; *Schoups et al., 2001*; *Ghose et al., 2002*; *Fritz et al., 2003*; *Yang and Maunsell, 2004*; *Blake et al., 2006*; *David et al., 2012*; *Xu et al., 2012*; *Goltstein et al., 2013*; *Jeanne et al., 2013*; *Poort et al., 2015*). Even in the absence of an explicit behavioral paradigm, repeated reward pairing can by itself already lead to an improvement of stimulus processing in sensory cortex (*Seitz et al., 2009*). However, it is unclear how visuospatial aspects of stimulus-outcome learning translate to changes in neuronal representations. The ecological relevance of this question is underscored by adaptive behaviors of rodents that differentiate between segments of visual space, such as when coping with potential predators moving overhead (e.g. birds of prey; *Zhang et al., 2012b*; *Shang et al., 2015*; *Wei et al., 2015*), or visual stimulus patterns at ground level, possibly associated with food and water (*Furtak et al., 2012*). Specifically, it is unknown how neural coding in the retinotopically organized visual cortex is affected by learning when adjacent locations in the visual field are differentially predictive of reward or other outcomes.

Located early in the visual stream, the primary visual cortex (V1) has neurons that primarily seem to act as low-level feature detectors, responding selectively to properties such as orientation, contrast, temporal and spatial frequency of bars and gratings, eye-specific inputs and retinotopic location (*Hubel and Wiesel, 1959*; *Hubel and Wiesel, 1962*; *Mrsic-Flogel et al., 2007*; *Niell and Stryker, 2008*; *Smith and Häusser, 2010*; *Freeman et al., 2013*). However, recent studies have

shown that activity in mouse V1 is strongly modulated by non-visual factors such as other sensory modalities, internal state, locomotion and anticipation of reward (*Shuler and Bear, 2006*; *Niell and Stryker, 2010*; *Bennett et al., 2013*; *Lee et al., 2014*; *Keller et al., 2012*; *Saleem et al., 2013*; *Meijer et al., 2017*). Widespread presence of feedback and neuromodulatory influences provide a means for learning experiences to drive plasticity and alter V1 local circuitry (*Gilbert, 1996*; *Chubykin et al., 2013*; *Zhang et al., 2014*; *Ji et al., 2015*).

Learning can affect neocortical function via a number of mechanisms. Neural responses in primary visual cortex may change to increase bottom-up saliency of a stimulus (*Zhang et al., 2012a*), for instance by a selective increase in amplitude of the response to conditioned stimuli (*Fritz et al., 2003*; *Blake et al., 2006*; *Andermann et al., 2010*; *Goltstein et al., 2013*; *Poort et al., 2015*). Alternatively, neurons can be recruited to become tuned to a conditioned stimulus (*Weinberger et al., 1993*), which in case of the tonotopic map of auditory cortex leads to an enlarged cortical region dedicated to this stimulus (*Bao et al., 2001*). Furthermore, the selectivity of neuronal tuning curves may be altered (*David et al., 2012*; *Goltstein et al., 2013*), for example amplifying small but relevant changes in stimulus properties to large changes in neuronal activity (*Schoups et al., 2001*). Finally, some changes can only be observed in simultaneously recorded neuronal populations, like changes in sparseness of population responses (*Ghose et al., 2002*; *Gdalyahu et al., 2012*) or the correlation structure of ensemble activity (*Averbeck et al., 2006*; *Poort and Roelfsema, 2009*; *Jeanne et al., 2013*; *Montijn et al., 2015*). While the effect of perceptual and reward learning in V1 is often restricted to smaller parts of the visual field (*Gilbert et al., 2009*; *Seitz et al., 2009*), less is known about how such plasticity unfolds with respect to the neuronal micro-architecture of V1's retinotopic organization.

Here, we probe how visuospatial stimulus-reward learning affects the spatial cortical organization of the conditioned stimulus representation. First, we chronically tracked the mesoscale neural representation of the reward-predictive field location within the retinotopic map. Next, we used two-photon calcium imaging to test whether visuospatial information was processed more reliably and efficiently by neurons located at the border between the cortical representations of the rewarded and non-rewarded stimulus.

## Results

### Head-restrained classical conditioning using spatially confined stimuli

Adult male C57Bl/6 mice were conditioned to associate moving gratings of identical orientation, but presented at different locations in the visual field, with upcoming reward or absence of reward (*Figure 1a,b,d* and *Figure 1—figure supplement 1a–e*). Animals were exposed to the conditioning paradigm for 10 to 17 daily sessions of 40 to 60 trials each (*Figure 1e*). In a conditioning paradigm such as presented here (comparable to trace conditioning), anticipatory nasal and oral movements (unconditioned response, for example licks and sniffs) can emerge in direct relation to the unconditioned stimulus (approach of the reward arm; *DeBold et al., 1965*; *Balsam, 1984*; *Huerta et al., 2000*; *Bensafi et al., 2003*; *Kehoe and Joscelyne, 2005*; *Drew et al., 2005*; *Joscelyne and Kehoe, 2007*; *Arzi et al., 2012*; *Raybuck and Lattal, 2014*). Therefore, we detected and quantified such movements offline (Online Materials and methods; *Figure 1c* and *Figure 1—figure supplement 2a*) and plotted them as a function of time relative to arrival of the reward arm in the final position.

From the first conditioning session onwards, putative licks and sniffs emerged aligned to stimulus onset (−6 s), offset (−2 s), reward arm movement onset (−0.5 s) and reward delivery (0 s; *Figure 1f*, left panel and *Figure 1—figure supplement 2b*). Initially there were no differences in behavioral responses between rewarded trials and non-rewarded trials, up to the time of reward consumption (Wilcoxon Matched-Pairs Signed-Rank (WMPSR) test; Nasal movement: p=0.32; Oral movement: p=0.90, n = 10 mice; *Figure 1f–g* and *Figure 1—figure supplement 2d*). From the second conditioning session onwards, however, a difference in anticipatory oral and nasal movements began to emerge between rewarded and non-rewarded trials in the period of reward-arm movement, defined from −0.4 s to −0.16 s before (non-) reward time (*Figure 1f* and *Figure 1—figure supplement 2c*). During this brief time window before reward delivery, no distinction between rewarded and non-rewarded trials could yet be made by anything other than the conditioned stimulus alone, as reward-arm movement trajectory and speed were identical in rewarded and non-rewarded trials up

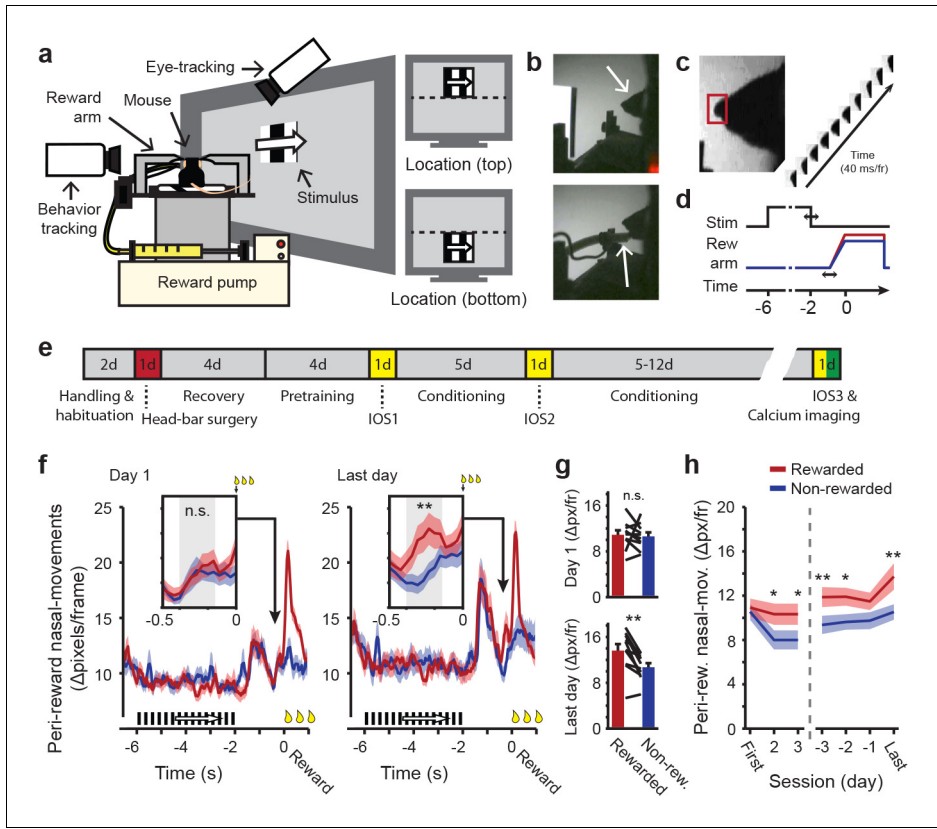

**Figure 1.** Classical conditioning in head-restrained mice. (**a**) Schematic depiction of the setup used for classical conditioning. Head-restrained mice faced a computer screen at a 45° angle. Reward (Vanilla dessert) was delivered through a tube on a movable arm (Reward arm). Cameras recorded putative sniffs and licks (Behavior tracking) and eye movements (Eye-tracking). Right: Example of compound stimuli used for conditioning; location ('top' or 'bottom') predicted reward, orientation of drifting gratings was identical in both locations during conditioning. (**b**) Outline of the mouse head (arrow, top panel). Reward-delivery (arrow, bottom panel). (**c**) Example of putative sniff, video-tracked in a small area surrounding the tip of the nose (red box). (**d**) Schematic showing the sequence of events in a single conditioning trial. Upper line shows stimulus onset and offset. Middle line represents reward arm position for rewarded (red) and non-rewarded (blue) trials. Lower line indicates time. Horizontal arrows indicate jitter in timing. (**e**) Timeline of the full conditioning experiment in days ('#d' indicates number of days; IOS1, IOS2 and IOS3 indicate intrinsic optical imaging time points; Red indicates day of head-bar implantation; Green indicates day of calcium imaging). (**f**) Peri-reward nasal movements in the first (left panel) and last (right panel) conditioning session. A large peak in nasal movements can be seen during reward delivery (t = 0) and in response to stimulus offset, 1.5 s before (non) reward. Reward-arm movement started at −0.4 s in both rewarded and non-rewarded trials. Insets: Nasal movements in the anticipatory period, when the arm was en-route, but did not exceed the non-reward position (gray box, −0.4 to −0.16 s before reward time). (**g**) Mean nasal movement in the anticipatory period in the first (top panel) and last (bottom panel) conditioning session. Single lines represent individual mice. (**h**) Mean anticipatory nasal movements on different days of the conditioning experiment. All panels: Data represent mean ± SEM across mice. Red lines: rewarded trials. Blue lines: non-rewarded trials. *p<0.05, **p<0.01, WMPSR test.

DOI: https://doi.org/10.7554/eLife.37683.002

The following figure supplements are available for figure 1:

**Figure supplement 1.** Operational procedures of the experimental apparatus for head-restrained conditioning.
DOI: https://doi.org/10.7554/eLife.37683.003

**Figure supplement 2.** Peri-reward oral-movement during visual conditioning.
DOI: https://doi.org/10.7554/eLife.37683.004

**Figure supplement 3.** Tracking of pupil diameter and movement during head-restrained conditioning of awake mice.
DOI: https://doi.org/10.7554/eLife.37683.005

to 55 ms before the time point at which reward was delivered or not (Online Materials and methods; *Figure 1—figure supplement 1f*). On the last day, all but one mouse showed increased nasal and oral movements during this reward-anticipation period (WMPSR test; Last day; Oral movements: p=0.004; Nasal movements: p=0.004, n = 10 mice; *Figure 1g,h* and *Figure 1—figure supplement 2e,f*). No differences in oral and nasal behaviors were found during the period of visual stimulation, which is not unexpected because reward was never available before the reward arm moved to the reward position. Therefore, the conditioning related increases in anticipatory nasal and oral movements during reward arm movement most likely reflect learned expectation of reward delivery.

## Eye movements during head-restrained conditioning

Awake behaving rodents display eye movements despite the absence of foveal vision (*Sakatani and Isa, 2004*; *Sakatani and Isa, 2007*; *Zoccolan et al., 2010*; *Adesnik et al., 2012*; *Keller et al., 2012*; *Wallace et al., 2013*; *Reimer et al., 2014*; *Vinck et al., 2015*). Although mice track the movement of bars at low temporal frequencies (<1 Hz; *van Alphen et al., 2010*), it is not known whether they preferentially orient their eyes towards behaviorally relevant stimuli. Large-amplitude eye movements could, in our experiments, deteriorate retinotopic selectivity of visual stimuli. Therefore, pupil location and pupil diameter were tracked in six animals over the entire course of the behavioral experiment (*Figure 1—figure supplement 3a*). This revealed a slight but systematic bias of pupil orientation towards the location of a presented stimulus along the vertical axis (Data combined across last 4 days, WMPSR test, Vertical: p=0.00091, Horizontal: p=0.39, n = 6 mice; *Figure 1—figure supplement 3b–g*). Nonetheless, the amplitude of the shift in vertical eye position did not differ by more than 1.0 retinal degrees (Online Materials and methods; *Figure 1—figure supplement 3c,d*). Given that V1 receptive fields in the mouse cover much larger regions (i.e. diameter $\geqq$10 retinal degrees; *Smith and Häusser, 2010*; *Bonin et al., 2011*) and that the conditioned stimulus size was 30 retinal degrees (Azimuth and Elevation), this subtle variation in eye position did not strongly reduce the retinotopic specificity of the visual stimulus during conditioning.

During the reward delivery period, eye movements and pupil dilations were more pronounced. The eye position showed a downward gaze-shift, potentially to the incoming reward spout (*Figure 1—figure supplement 3c*). Dilation of the pupil may have indicated arousal as a consequence of reward consumption (Data combined across last 4 days: WMPSR test, p=$7.1\cdot10^{-5}$, n = 6 mice; *Figure 1—figure supplement 3i,j*; *Bradley et al., 2008*). Thus, despite the lack of foveal vision, head restrained mice make eye-movement responses and show pupillary reactions to behaviorally relevant stimuli, like rewards. Differences in these behaviors between rewarded and non-rewarded trials were large during the reward delivery phase, while during stimulus presentation differences between the upper and lower visual field were negligible.

## Mesoscopic shifts in cortical representations of trained stimuli

To follow the cortical retinotopic organization throughout learning, we used repeated transcranial imaging of intrinsic optical signals in area V1 of anesthetized mice. The retinotopic pattern of the mesoscale cortical response was quantified by reducing the spatial dimensionality of the intrinsic response maps to the axis that maximally separated the two stimulus locations (Online Materials and methods; *Figure 2a–b* and *Figure 2—figure supplement 1a*). To account for variation between sessions, the response to trained stimuli was referenced to the response to control stimuli that were presented in the same retinotopic location (and had an orientation that was not presented in the conditioning sessions; for example gray curves in *Figure 2d* and *Figure 2—figure supplement 1b*). Using this method, we observed that the response amplitude to the non-rewarded, trained stimulus was decreased after conditioning when compared to before conditioning (amplitude of non-rewarded minus orthogonal, across time points: Kruskal-Wallis, normalized per mouse, $H_{(2, 24)}$=13.15, p=0.0014, post hoc WMPSR test, Before vs. After, p=0.0039; n = 9 mice; *Figure 2d,g* and *Figure 2—figure supplement 1b*). The response amplitude to the rewarded conditioned stimulus was not significantly reduced at the time point after conditioning (amplitude of rewarded minus orthogonal, mouse-normalized, across time points: Kruskal-Wallis, normalized per mouse, $H_{(2, 24)}$=6.39, p=0.041, post hoc WMPSR test, Before vs. After, p=0.57, n = 9 mice).

The change in response amplitude was associated with an altered spatial distribution of the intrinsic signal response to the trained stimuli (see *Figure 2d*). In order to further quantify this, we

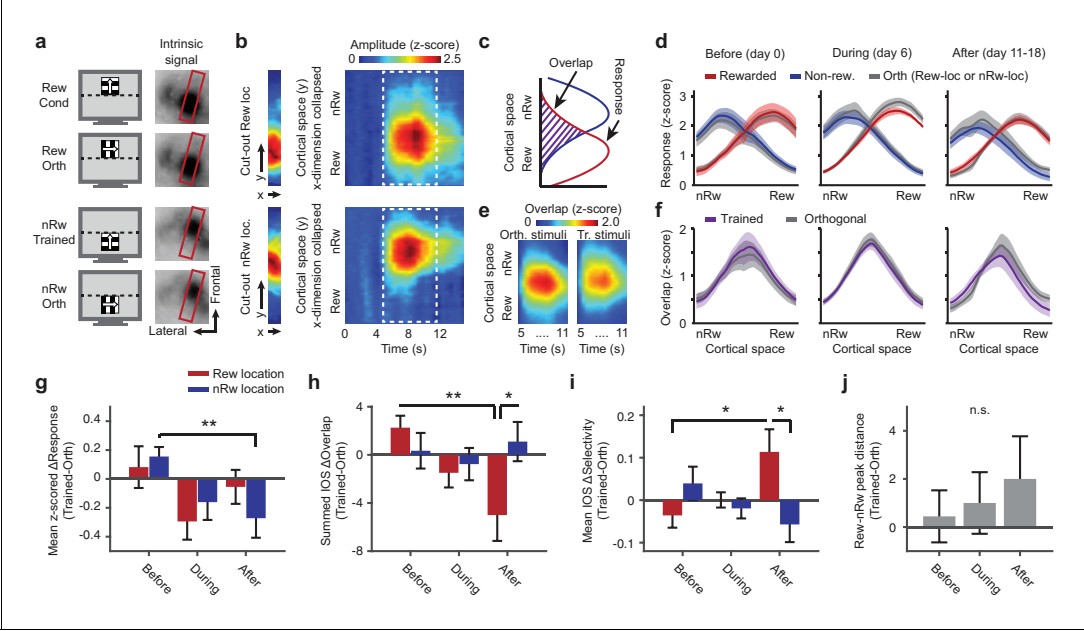

**Figure 2.** Repeated imaging of intrinsic optical signals in V1 of mice subjected to conditioning. (a) Visual stimuli (left panels) and average stimulus-induced intrinsic optical signal (IOS; right panels), showing strong retinotopically selective responses to the rewarded and non-rewarded location in area V1 (within red boundary) and weaker responses in the lateral supplementary visual areas LM and AL (to left of red box). Orth, orientation orthogonal to trained orientation (Cond). (b) Left panels: Example of 11-pixel wide cutout from the red box in a, that was automatically rotated so as to include the cortical area of maximal activation for both the top and bottom stimulus location. Right panels: The IOS response to each stimulus as a function of cortical space (maximally separating the top and bottom stimulus location) and time (derived from the 11-pixel wide cutouts by collapsing the 'x' dimension per imaging frame and concatenating time points). Upper panels: Response to rewarded location and orientation. Lower panels: Response to non-rewarded location and orientation. (c) Schematic showing quantification of the response amplitude and overlap of cortical responses to visual stimuli in adjacent retinotopic locations. (d) Mean ± SEM z-scored amplitude of the IOS response (across the period of maximum activation, 7 to 11 s) as a function of cortical space along the axis that maximally separates the rewarded and non-rewarded stimulus. Red: Rewarded stimulus; Blue: Non-rewarded, trained stimulus; Grey: Orthogonal orientations at the rewarded or non-rewarded location respectively. Left panel: Data acquired before the first conditioning session (day 0; IOS1, see also *Figure 1e*). Middle panel: After five conditioning sessions (IOS2). Right panel: Data acquired after the last conditioning session on the same day as, but before, calcium imaging (IOS3). (e) Mean overlap (across all mice) between the response to the orthogonal or trained stimuli in the rewarded and non-rewarded locations as a function of cortical space (y-axis) and time (x-axis). Scale bar above applies for both panels. (f) As in d), but for the overlap between the response to the rewarded and non-rewarded trained stimulus (plotted in Violet). (g) ΔResponse amplitude (difference between trained and control orientations) in the cortical region that responded to the rewarded location (red) and non-rewarded location (blue) separately, across imaging time points (**p=0.0039, Kruskal-Wallis, post hoc WPMSR test). (h) As in g), but for ΔOverlap per region and across imaging time points (*p=0.039, WMPSR test; **p=0.011, Kruskal-Wallis test, post hoc WMPSR test). (i) As in g), but for ΔSpatial selectivity (Rewarded vs. Non-rewarded: *p=0.039, WMPSR test; Before vs. After: *p=0.046, Kruskal-Wallis test). (j) Distance between the peak responses to the spatially segregated stimuli (difference between trained and orthogonal control stimuli, positive values indicate larger distance between trained stimulus representation peaks).

DOI: https://doi.org/10.7554/eLife.37683.006

The following figure supplement is available for figure 2:

**Figure supplement 1.** Repeated intrinsic optical signal imaging in V1 of a mouse subjected to conditioning.

DOI: https://doi.org/10.7554/eLife.37683.007

calculated how much the intrinsic response to visual stimulation in each of the two adjacent locations of the trained stimuli overlapped with the other on the cortical surface (*Figure 2c,e*) and compared this with how much the response to the orthogonal control stimuli overlapped (difference between gray and colored lines in *Figure 2f*). This difference between spatial overlap of trained and spatial overlap of control stimuli (IOS ΔOverlap) inversely reflects how well the intrinsic responses to trained stimuli are segregated in cortical space, as compared to the control stimuli. After conditioning (i.e., on day 11–18), the IOS ΔOverlap was significantly reduced in the cortical region that responded to the rewarded stimulus (Kruskal-Wallis test, normalized per mouse, $H_{(2,24)} = 10.00$, p=0.0067, post hoc WMPSR test, Before vs. After, p=0.020, n = 9 mice; *Figure 2f,h*). In the region responding to the non-rewarded stimulus, the IOS ΔOverlap was not significantly reduced as a function of training

(Kruskal-Wallis test, normalized per mouse, $H_{(2,24)}$ = 0.98, p=0.61, n = 9 mice; *Figure 2f,h*). After training, the IOS ΔOverlap in the cortical region representing the rewarded stimulus was significantly reduced compared to the region mostly responsive to the non-rewarded stimulus (WSRMP test, p=0.039, n = 9 mice; *Figure 2f,h*).

Because the measure of overlap is sensitive to the absolute response amplitude, we also quantified the mesoscopic segregation of intrinsic cortical responses using a ratiometric index, the spatial selectivity index. This showed a similar effect of sharpened spatial selectivity for the rewarded condition and location (Kruskal-Wallis test, mouse-normalized, $H_{(2,24)}$ = 6.16, p=0.046, WMPSR test, rewarded vs. non-rewarded, p=0.039, n = 9 mice; *Figure 2i*). The magnitude of these above-mentioned effects ('IOS-ΔOverlap', 'selectivity for stimulus location' and 'response amplitude to the non-rewarded trained stimulus') correlated strongly within individual mice (IOS-ΔOverlap × Selectivity: r = −0.94, p=0.0002, n = 9 mice; Amplitude × Selectivity: r = 0.81, p=0.008, n = 9 mice; Amplitude × ΔOverlap: r = −0.65, p=0.058, n = 9 mice), which suggests that they resulted from closely related mechanisms that operated on the rewarded as well as the non-rewarded side of the stimulus representation.

Finally, to test whether the representation of the rewarded and non-rewarded trained stimuli drifted apart in cortical space, we quantified the distance between the peaks in the intrinsic response profiles. Despite a trend, this measure of spatial distance between trained stimulus representations did not significantly increase (Kruskal-Wallis test, normalized per mouse, $H_{(2,24)}$ = 0.57, p=0.75, n = 9 mice; *Figure 2j*). Hence, classical conditioning, using rewarded and non-rewarded stimuli that occupy neighboring regions of visual space, selectively reshaped the retinotopic organization for the trained stimuli compared to control stimuli, while the overall representations of the trained stimuli did not significantly drift apart.

## Spatial organization of neuronal population activity for trained stimuli

To understand how formation of a visuospatial stimulus-reward association affects neuronal response properties within the retinotopic representation, we performed OGB1-calcium imaging (*Stosiek et al., 2003*) after the final conditioning session (*Figure 3a*). Recordings were made in multiple V1-subregions per animal, positioned along the rewarded/non-rewarded axis within V1 (*Figure 3b*). Every field-of-view was assigned to a cortical location group; Full non-rewarded, Border non-rewarded, Border rewarded and Full rewarded, based on the population response amplitude to visual stimulation in each of the retinotopic locations using moving gratings of non-trained orientations (Online Materials and methods; *Figure 3b* and *Figure 3—figure supplement 1*).

Using this location on the rewarded/non-rewarded axis, we compared findings obtained using intrinsic signal imaging with calcium imaging data. First, we calculated the measure $Ca^{2+}$ ΔOverlap, which was defined as the overlap between the rewarded- and non-rewarded stimulus-evoked calcium response, using mean cellular calcium responses to the trained orientations for each imaging field. These overlap values were averaged across imaging fields per cortical location bin and subtracted by the (identically calculated) overlap of the orthogonal control stimuli. The $Ca^{2+}$ ΔOverlap was significantly lower for imaging fields in the full rewarded region and at the rewarded/non-rewarded location border, compared to field-of-views that were located fully in the non-rewarded stimulus region (Kruskal-Wallis test, $H_{(3,45)}$ = 11.6, p=0.0088, n = 49 imaging fields; *Figure 3c*). The pattern of overlap in location selective responses based on calcium imaging data resembled the results obtained using intrinsic signal imaging (Gray line in *Figure 3c* indicates IOS ΔOverlap as in *Figure 2d*), showing the strongest reduction in Overlap (as compared to control) in the Border rewarded and Border non-rewarded regions. Also selectivity for retinotopic location, now calculated using calcium imaging data, was larger in the Full rewarded and Border rewarded regions, as compared to the Full non-rewarded region (Kruskal-Wallis test, $H_{(3,45)}$ = 19.3, p=0.0024, n = 49 imaging fields). The measured ΔOverlap in the rewarded stimulus location correlated strongly and significantly across methods (IOS vs. $Ca^{2+}$, r = 0.81, p=0.016, n = 8 mice) and a weaker, but similarly positive correlation was found for retinotopic location selectivity (r = 0.69, p=0.058, n = 8 mice).

If the spatial segregation of the cortical population responses was driven by local map expansion, this would predict a larger fraction of neurons being tuned to rewarded or non-rewarded stimulus orientations compared to orthogonal orientations. The fraction of responsive neurons per stimulus orientation and location can be estimated only from the calcium imaging data acquired at single cell resolution and cannot be directly compared to intrinsic imaging data. Interestingly, the fraction of

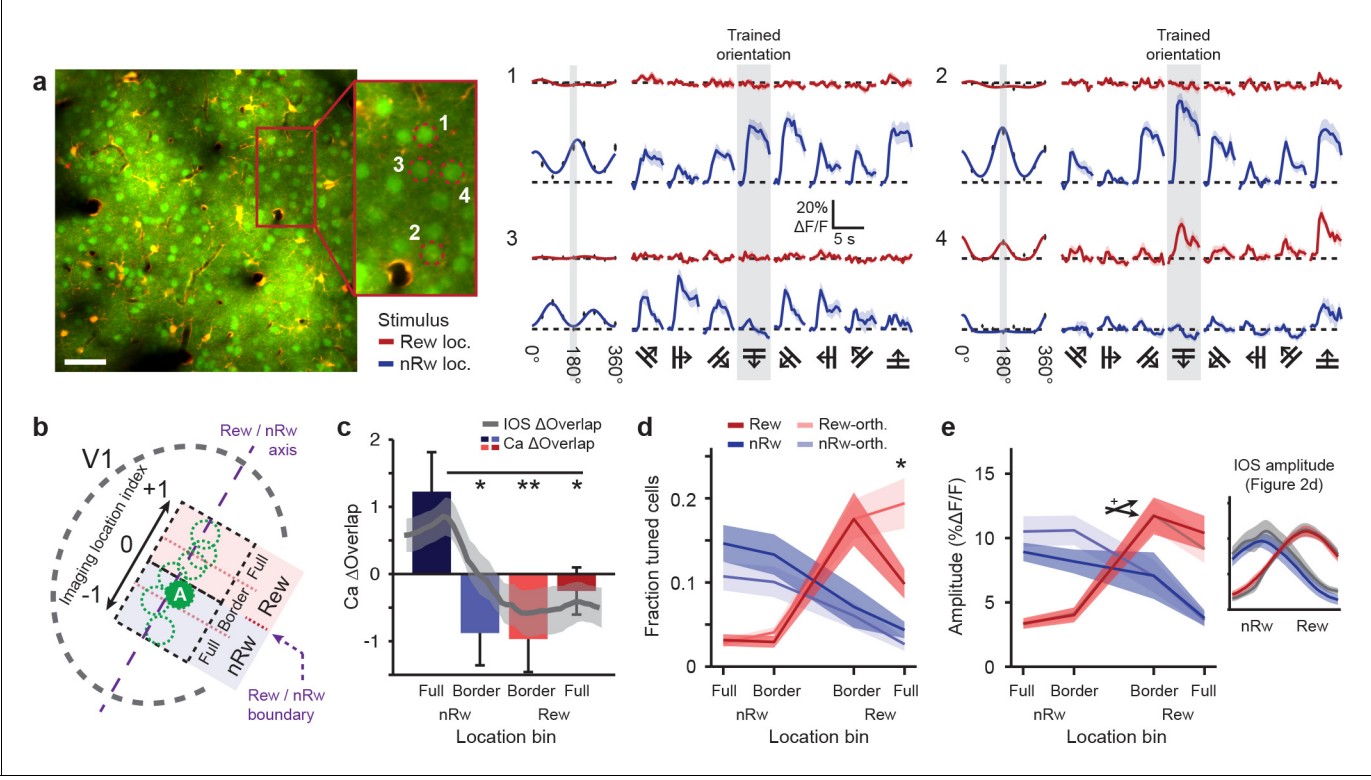

**Figure 3.** Two-photon imaging of orientation-tuned responses in different retinotopic regions of V1. (**a**) Example of calcium imaging in a field of view that was located in the Border non-rewarded region. Left: Overview image and inset showing neurons loaded with the calcium indicator OGB1-AM (green) and double-labeled astrocytes (yellow). Right panels 1–4 correspond to ROI's 1–4 in inset a. Left columns of all 4 ROI response panels: Fitted tuning curves for movement orientation/direction (Red: visual stimulus in rewarded visual field; Blue: visual stimulus in non-rewarded visual field) and mean (±SEM) response to each of the eight directions (black vertical bars). Traces on right: Average ΔF/F time courses for each of the eight movement directions separately (Grey shaded columns: trained orientation). (**b**) Top-view scheme of the imaging locations on the visual cortex. The dashed purple line represents the 'rewarded/non-rewarded axis'. The boundary between the rewarded and non-rewarded stimulus representations is at location index value '0' and marked with a dashed purple arrow (see Online Materials and methods). 'A' indicates the approximate location of the field of view in a. (**c**) Calcium-imaging derived ΔOverlap in each of the location bins. Overlaid gray line represents mean (±SEM, shaded region) of the overlap measured after conditioning using intrinsic imaging (*p<0.05, **p<0.01, Kruskal-Wallis, post-hoc Mann-Whitney U test). This overlap was computed as in *Figure 2*. (**d**) Fraction of orientation-tuned neurons preferring the rewarded (red), non-rewarded (blue) or orthogonal (lighter shaded colors) stimuli (*p<0.05, Anova, post-hoc t-test). (**e**) Average response amplitude in ΔF/F (%) of significantly orientation-tuned neurons to their respective preferred orientations, per location bin. Red: Cells preferring the rewarded orientation. Light red: Cells preferring the orthogonal stimulus in the rewarded location. Blue and light blue: The same, but for the non-rewarded stimulus location. Crossed arrow indicates Anova interaction effect,+P < 0.05. Inset on the right shows the mean (±SEM) response amplitude as measured in the last intrinsic imaging session (see *Figure 2d*).

DOI: https://doi.org/10.7554/eLife.37683.008

The following figure supplement is available for figure 3:

**Figure supplement 1.** Binning of recording location into discrete groups.

DOI: https://doi.org/10.7554/eLife.37683.009

responsive cells per stimulus orientation was significantly reduced in the Full rewarded location and did not differ significantly between the Border rewarded and Border non-rewarded locations (2-way Anova, interaction effect of Location bin versus Preferred orientation, $F_{(3,90)} = 3.99$, p=0.010; n = 49 recordings; Post hoc, Trained versus Orthogonal orientation for full-rewarded location: p<0.05; *Figure 3d*). Instead, we observed that the population response amplitude to rewarded stimuli increased more steeply from Border non-rewarded locations to Border rewarded locations compared to the increase of the population response to non-rewarded stimuli from Border rewarded to Border non-rewarded locations (2-way Anova, interaction effect of location vs. population response difference, Trained stimuli: $F_{(3,85)} = 3.59$, p=0.017, Untrained stimuli: $F_{(3,85)} = 1.48$, p=0.23; n = 49 recordings; *Figure 3e*). While the response amplitude to the non-rewarded trained orientation appeared also smaller compared to the non-rewarded control orientation, similar to what we

observed in the intrinsic imaging data (see *Figure 3e*, inset right), this difference was not significant. Nonetheless, the reduction in response amplitude (trained orientation minus orthogonal control), as obtained using calcium imaging, did covary with the reduction in response amplitude that was observed in the intrinsic imaging data (IOS vs. Ca$^{2+}$, r = 0.92, p=0.028, n = 5 mice for which we had both intrinsic imaging data and calcium imaging recordings from the non-rewarded location).

These results suggest that the observed mesoscopic disambiguation of the representation of the rewarded and non-rewarded stimulus (as observed using intrinsic imaging) is expressed at the single neuron level (as observed using calcium imaging) and is potentially mediated by a spatially restricted change in response patterns of a select population of neurons, rather than an overall expansion of the cortical map for rewarded or non-rewarded stimuli.

## The rewarded conditioned stimulus drives a smaller population of orientation-tuned neurons with larger response amplitudes

To investigate how response patterns of orientation-tuned neurons were altered after conditioning, we quantified tuning curve parameters for individual neurons as a function of their imaging location (Full vs Border, with rewarded/non-rewarded pooled), whether the cell responded to the rewarded or non-rewarded stimulus location (rewarded vs non-rewarded, with Full/Border pooled), and the similarity of the neuron's preferred orientation to the trained orientation. While the populations of orientation-tuned cells located in Border- and Full imaging regions did not show significant differences in fraction of neurons being tuned to trained and other orientations, response amplitude and tuning curve bandwidth (Kruskal-Wallis test, p>0.05; n = 11 mice; *Figure 4—figure supplement 1*), we observed significant differences between these parameters for cells that responded to the rewarded and non-rewarded stimulus location.

On average, the percentage of cells that preferred the rewarded and non-rewarded stimulus location was similar, although not identical (total: 32.2%, see *Table 1* for fraction per mouse; rewarded: 17.8%, non-rewarded: 14.5%; WMPSR-test, p=0.58, n = 11 mice; *Figure 4a*). However, the relative fractions of cells preferring trained and other orientations differed significantly between groups that were tuned to rewarded and non-rewarded locations (these fractions were normalized to the overall fraction of responsive neurons per mouse, across all preferred orientations and locations and adjusted for relative bin width so that the expected fraction given a flat distribution of preferred orientations was 1.0; *Figure 4b,c*). Specifically, for cells tuned to the rewarded location there was a relatively smaller fraction of cells preferring the trained orientation, compared to the set of cells that responded to the non-rewarded (i.e. orthogonal) orientation (Kruskal-Wallis test, 'Rewarded−Non-rewarded' vs. 'ΔOri. from trained', $H_{(4,50)}$ = 9.76, p=0.045; post hoc WMPSR-test, '0°–10°' vs. '50°–70°', p=0.042, n = 11 mice; *Figure 4d*). This effect corresponds to what is shown in *Figure 3d*, in which we observed fewer neurons being responsive to the trained stimulus in the Full rewarded cortical region compared to the orthogonal control orientation.

Although the fraction of cells tuned to the conditioned orientation at the rewarded stimulus location was reduced, their response amplitude to the trained orientation was increased compared to cells responding to the non-rewarded location (Kruskal-Wallis test, 'Rewarded−Non-rewarded' vs. 'ΔOri. from trained', $H_{(4,48)}$ = 11.6, p=0.020; n = 11 mice; *Figure 4e*). Moreover, neurons with tuning curves that flanked the preferred orientation (ΔOri. from trained between 10° and 30°) had broader tuning curves (Kruskal-Wallis test, 'ΔOri. from trained', $H_{(4,48)}$ = 14.1, p=0.0068; Post hoc WMPSR test, '0°–10°' vs. '10°–30°', p=0.0039; n = 11 mice; *Figure 4f*). The latter effect indicates that neurons with preferred orientations that slightly differ from the trained orientation increased their response to the trained orientation (regardless of the reward-location association), rendering their orientation tuning curves broader (*Goltstein et al., 2013*). The orientation selectivity index did not significantly differ between rewarded and non-rewarded tuned neurons, or as function of angle between preferred orientation and the trained orientation (Kruskal-Wallis test, p>0.16; n = 11 mice). When we performed these analyses on tuning curves of the neurons that were not significantly tuned, we did not observe any of the above-described differences.

In order to test whether the differences described above reflected an overall sparser population response to rewarded conditioned stimuli, we quantified population sparseness (a$^{p}$, lower values indicate a sparser population response; *Rolls and Treves, 2011*) for the tuning curve response of all neurons. The population response to trained stimuli in the rewarded location did not show a significantly lower sparseness compared to the population response to visual stimulation in the non-

**Table 1.** Overview of mice included in behavioral and imaging analyses

Each row lists the following information for the mouse that can be identified by the number in the column Mouse. Rew: Location of the rewarded stimulus. nRw: Location of the non-rewarded stimulus. Dir: Direction of the moving grating used in the conditioning paradigm. Beh-track: Data for which behavior was video-tracked and included in behavioral analysis (*Figure 1* and *Figure 1—figure supplement 2*). # trials: Total number of trials that a mouse performed across all conditioning sessions. Eye: Data included in analysis of eye-movements (*Figure 1—figure supplement 3*). IOS: Data included in analysis of intrinsic optical signals (*Figure 2* and *Figure 2—figure supplement 1*). Ca²⁺ Full: Data included in calcium imaging analysis, 'Full imaging locations'. Ca²⁺ Border: Data included in calcium imaging analysis, 'Border imaging locations'. (*Figures 3–6*, *Figure 3—figure supplement 1*, *Figure 4—figure supplement 1*). % tuned: Overall fraction of orientation tuned neurons per mouse.

| Mouse | Rew | nRw | Dir | Beh-track | # trials | Eye | IOS | Ca²⁺ Full | Ca²⁺ Border | % tuned |
|---|---|---|---|---|---|---|---|---|---|---|
| #02 | Bottom | Top | 180° | | | | | * | * | 30.9 |
| #03 | Top | Bottom | 0° | | | | | * | * | 10.9 |
| #05 | Bottom | Top | 0° | | | | | * | * | 68.8 |
| #07 | Top | Bottom | 90° | * | 613 | | * | * | * | 20.2 |
| #08 | Top | Bottom | 270° | * | 378 | | * | | | |
| #09 | Bottom | Top | 180° | * | 640 | | * | * | * | 36.9 |
| #10 | Bottom | Top | 180° | * | 575 | | * | * | | 40.6 |
| #11 | Top | Bottom | 180° | * | 436 | * | * | * | * | 32.8 |
| #12 | Top | Bottom | 0° | * | 520 | * | * | * | * | 36.9 |
| #13 | Bottom | Top | 90° | * | 595 | * | | | | |
| #14 | Bottom | Top | 270° | * | 640 | * | * | * | * | 40.7 |
| #15 | Top | Bottom | 0° | * | 520 | * | * | * | * | 40.5 |
| #16 | Bottom | Top | 0° | * | 480 | * | * | * | | 7.2 |
| Total | | | | 10 | | 6 | 9 | 11 | 9 | |

DOI: https://doi.org/10.7554/eLife.37683.012

rewarded location (Kruskal-Wallis test, 'ΔOri. from trained', $H_{(4,50)}$ = 0.23, p=0.99; n = 11 mice; *Figure 4g*).

These differences in response properties of orientation-tuned neurons indicate that, although fewer neurons were significantly responsive to the conditioned stimulus orientation, the individual neurons that responded to the rewarded stimulus did so more strongly.

## Improved neuronal population coding for trained stimulus location

Thus far our data obtained using intrinsic signal imaging as well as calcium imaging indicate that, after training, the cortical map for trained stimulus location and orientation became more selective for retinotopic position. We therefore hypothesized that populations of simultaneously recorded single neurons would show an improved ability to discriminate the location of trained stimuli as compared to the (same) locations of untrained control stimuli. To test this, we used a Bayesian algorithm with cross-validation to decode stimulus location from single trial responses to the very stimuli that were used in the conditioning paradigm. Performance of the decoding algorithm for those stimuli was compared with that for stimuli having orientations differing 45 or 90 degrees from the trained orientation (untrained stimuli).

First, we decoded calcium responses recorded in field-of-views located around the rewarded/non-rewarded border on the cortical surface (see *Figures 3b* and *5a*, inset left). In this population of significantly tuned Border neurons, decoding of stimulus location was significantly better for the trained orientation as compared to the orthogonal orientation (WMPSR test, difference between 'trained' and 'orthogonal orientation' averaged across all sample sizes per mouse, p=0.0078; n = 9 mice; *Figure 5a*, left panel). Decoding of activity patterns from cells recorded in regions further away from the border (see *Figure 5a*, right panel) did not differ between trained and untrained

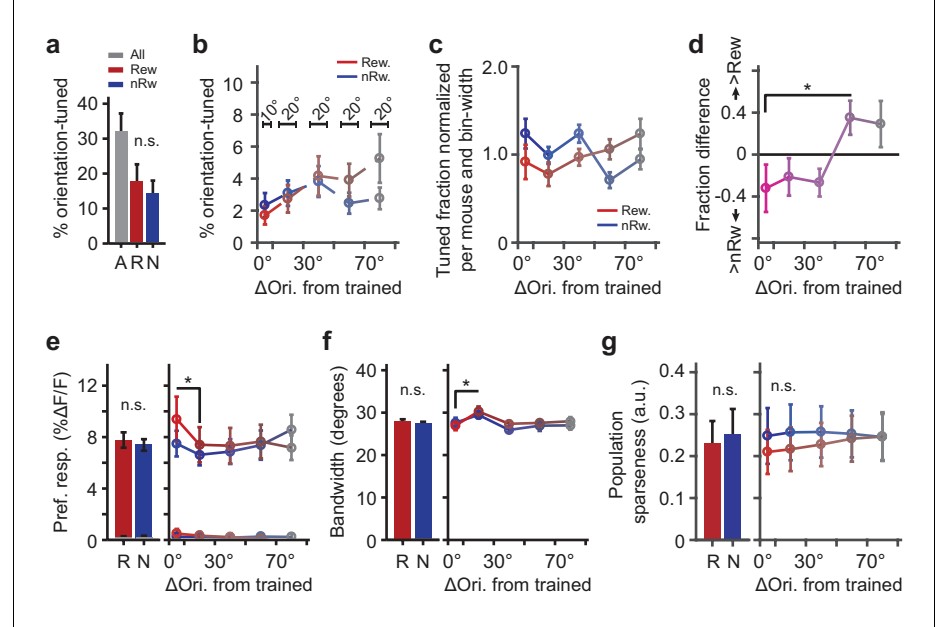

**Figure 4.** Parameters describing orientation-tuned neurons preferring the rewarded or non-rewarded stimulus location. (**a**) Left panel: Mean ± SEM percentage of significantly orientation-tuned cells per mouse. 'A': All locations. 'R': Rewarded location. 'N': Non-rewarded location. (**b**) Absolute fraction of orientation-tuned neurons responding to the rewarded (red) and non-rewarded (blue) location separately, and relative to the trained orientation (e.g. neurons with a preferred orientation nearly identical to the trained orientation were in bin 0°−10°). Bars above data points indicate relative bin-widths. Note that because the left-most bin (0–10 degrees) is only half the width of the other bins, it is expected to contain half of the fraction of the other bins. (**c**) Distribution (normalized per mouse and bin-width) of preferred orientations relative to the trained orientation (given a flat distribution of preferred orientations, the expected fraction of each bin would be 1.0). (**d**) Difference in fraction of orientation tuned neurons to the rewarded minus the non-rewarded location (positive values indicate a larger fraction of neurons tuned to the rewarded stimulus location; *p=0.042, Kruskal-Wallis test, post hoc WMPSR test). (**e**) The mean (±SEM) ΔF/F response to the preferred direction as a function of the preferred orientation relative to the trained orientation for rewarded and non-rewarded location-tuned neurons separately (*p<0.05, WMPSR test). The lines that fall between 0% to 0.5% ΔF/F show the response amplitude of the same neurons, but to the orientation that was orthogonal to the preferred orientation. We note that the sequence of bins is thus not representative of the entire tuning curve but shows the response amplitude to a specific (preferred or orthogonal) direction. (**f**) Same as **e**), but for bandwidth (*p<0.05, WMPSR test). (**g**) As in **e**), but for sparseness of the tuning curve response calculated across the population all neurons. For all panels: Red colors indicate groups of cells preferring 'rewarded' conditioned orientations, while blue colors indicate groups of cells preferring 'non-rewarded' trained orientations. Both colors fade to gray, indicating that the preferred orientation of the cells becomes more dissimilar from the trained orientation.

DOI: https://doi.org/10.7554/eLife.37683.010

The following figure supplement is available for figure 4:

**Figure supplement 1.** Comparison of orientation tuning parameters between cells located close and far away from the rewarded/non-rewarded border.

DOI: https://doi.org/10.7554/eLife.37683.011

orientations (WMPSR test, difference between 'trained' and 'orthogonal orientation' averaged across all sample sizes per mouse, p=0.46; n = 11 mice; *Figure 5a*, right panel).

Data from six separate animals contributed both behavioral measurements and Border-region imaging fields of view. Of these, five mice showed better decoding of trained stimulus location compared to untrained orientations, while a single animal did not show a difference. Moreover, the magnitude of the difference in decoding performance between trained and other (45° and 90°) orientations correlated significantly with the magnitude of the difference in anticipatory nasal and oral movements in the behavioral paradigm (Nasal movements: r = 0.83, p=0.041; Oral movements: r = 0.96, p=0.003; n = 6 mice; *Figure 5b*). Decoding performance did not correlate significantly with

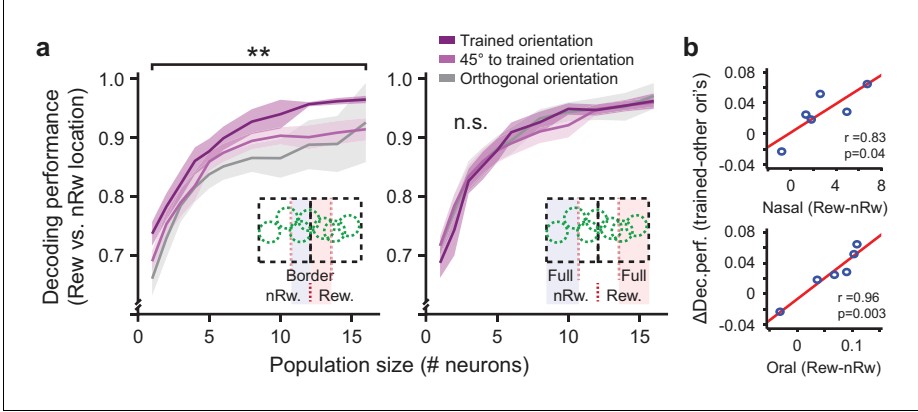

**Figure 5.** Population coding of visual field location by visual cortex neurons. (a) Left panel: Performance of decoding retinotopic location using activity of simultaneously recorded neurons in regions close to the rewarded/non-rewarded location border as a function of increasing population size. Chance level is 50%. Inset illustrates which field-of-view locations were included in these data. Dark purple curves are data from the population with a preferred orientation similar to the trained orientation. Light purple curves are data from neurons preferring orientations differing 45° from the trained orientation. Gray curves are for the data from cells preferring orientations orthogonal to the trained orientation. Data are shown as mean ± SEM averaged across 9 mice. **p<0.01, WMPSR test. Right panel: Same, but for neurons recorded in regions further away from the rewarded/non-rewarded location border. (b) Per mouse, the mean conditioning related-improvement in coding for visual field location was plotted against the behavioral difference in nasal (upper panel) and oral movement (lower panel) to the rewarded (Rew) and non-rewarded (nRw) stimulus location.

DOI: https://doi.org/10.7554/eLife.37683.013

the total number of conditioning trials that the mice were exposed to (Total number of trials: r = −0.18, p=0.73; n = 6 mice). The single mouse that did not show a significant improvement in

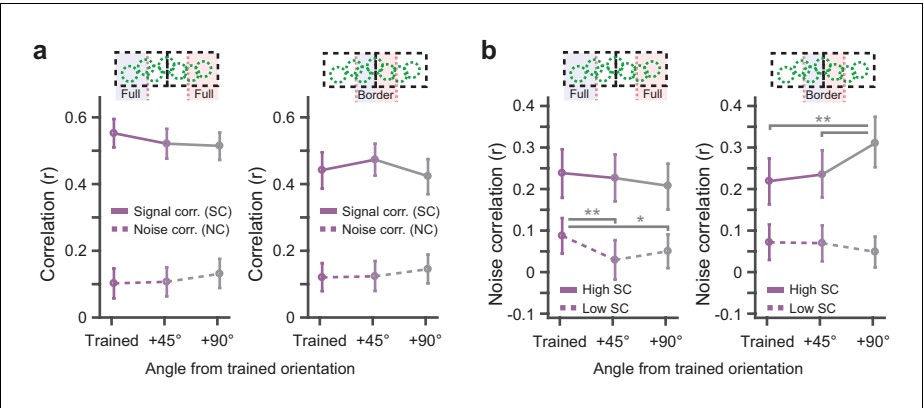

**Figure 6.** Conditioning related interactions of signal and noise correlations. (a) Signal- and noise correlations for pairs of simultaneously recorded neurons as a function of the difference between the preferred orientation of the neurons and the trained orientation. Left panel: Data from pairs that were located further away from the region where the rewarded and non-rewarded stimuli bordered. Right panel: As left, but from pairs located in the cortical region where the representations of the rewarded and non-rewarded stimuli bordered. Solid lines: Signal correlations. Dashed lines: Noise correlations. Data are shown as mean ± error bars that indicate 95% confidence intervals. (b) Noise correlations for the selection of simultaneously recorded pairs that exhibited the 10% highest signal correlations (solid lines) and the pairs having the 10% lowest signal correlations (dashed lines). Left panel: data from the Full rewarded and Full non-rewarded regions. Right panel: data from Border rewarded and non-rewarded regions. Data are shown as mean ± error bars that indicate 95% confidence intervals (*p<0.05, **p<0.01 multiple comparison corrected bootstrap confidence intervals).

DOI: https://doi.org/10.7554/eLife.37683.014

decoding performance also failed to show an increase of both anticipatory nasal- and oral-movements in the behavioral paradigm, while it was still exposed to 640 conditioning trials (see *Table 1*, mouse #09).

## Conditioning-related suppression of noise correlations in high signal correlation pairs

Effects of conditioning were observed in a subset of similarly tuned neurons within the overall heterogeneously tuned group of V1 neurons. We assessed the degree to which cells in these subgroups can be seen as functional units by comparing similarity of tuning curves (signal correlation) and stimulus-independent covariations (noise correlation; see for example *Averbeck et al., 2006*; *Montijn et al., 2016*). Signal and noise correlations were calculated on tuning curves of all simultaneously recorded pairs of cells that were either responsive to the rewarded or non-rewarded location and grouped according to the similarity of their preferred orientations relative to the trained orientation, and whether they were located in the Full or Border stimulus region. However, there were no significant differences in average signal correlations and noise correlations between the trained and control orientations (Bootstrap confidence intervals, p>0.05; *Figure 6a*).

Interactions between signal and noise correlations can hamper or facilitate efficient population coding. For instance, population coding benefits when cells with high signal correlations have low noise correlations and vice versa (*Oram et al., 1998*; *Averbeck et al., 2006*; *Poort and Roelfsema, 2009*; *Jeanne et al., 2013*). To investigate whether such an effect occurred in our data, we selected pairs of cells that displayed the 10% highest and 10% lowest signal correlations and calculated noise correlations for these pairs separately (*Jeanne et al., 2013*). In the border imaging regions, high signal correlation pairs that were tuned to the trained orientations exhibited significantly lower noise correlations compared to pairs preferring other orientations (Bootstrap confidence intervals, p<0.01; $n_{border}$ = 20 imaging fields of view; *Figure 6b*, right panel, solid lines). In addition, in the Full imaging regions, we observed that low signal correlation pairs had stronger noise correlations when they were tuned to the trained orientations as compared to the control orientations (Bootstrap confidence intervals, p<0.05; $n_{full}$ = 24 imaging fields of view; *Figure 6b*, left panel, dotted lines). Thus, the effect of classical conditioning on noise correlations depends on the signal correlation of the pair of neurons, which can have the overall effect of enhancing the discriminability of trained stimuli in the population.

## Discussion

Associative learning is known to affect neural responses across a range of spatial scales (*Jehee et al., 2012*; *Bao et al., 2001*; *Goltstein et al., 2013*), but how learning affects the spatial dimension of sensory cortical maps has remained elusive. Our results indicate how associative learning (by coupling a visual stimulus to reward) affects orientation-specific coding of single neurons in a retinotopically restricted fashion but fails to lead to an overall expansion of the cortical region representing the trained visual space, thus leaving the retinotopic map intact. These data demonstrate that learning-induced changes in single neurons emerge in concert with enhanced stimulus representations at the population level increasing location discriminability within the V1 cortical map.

### Considerations regarding imaging methodology

The central finding in this study is improved coding for stimulus location by a population of neurons that preferred the trained orientation *and* was located within the cortical retinotopic region where the trained stimuli bordered. This enhanced selectivity for retinotopic location was observed using both a mesoscale (intrinsic imaging) and a microscale (calcium imaging) method. The data acquired using intrinsic imaging suggested that this effect was mediated by a reduction in response amplitude to the non-rewarded trained stimulus (*Figure 2d,g*). While visibly present, this reduction was less pronounced when observed at the single cell resolution using calcium imaging (*Figure 3e*). The difference in results may be inherent to the difference in methods, with intrinsic signal imaging picking up a widespread mixed, metabolic signal across cortical neurons (including dendrites, axons and synapses) and layers, and with calcium imaging focusing on somatic spiking activity in layer 2/3 neurons selectively. That the magnitude of reduction in trained stimulus-representation Overlap, as measured with each method (*Figure 2f* and *Figure 3c*), was correlated across mice suggests that the effects

are related. What was observed at the mesoscale as an overall change in response amplitude, may in part reflect a subtler change in stimulus selectivity or jointly orchestrated activity patterns at the single neuron, somatic level within cortical layer 2/3.

It is of note that the results obtained with calcium imaging, such as differences in fraction and amplitude of responsive neurons and changes in population decoding for retinotopic location, were observed in comparison to untrained control orientations or control regions, and not compared to a hypothetical time point before learning. Therefore, it cannot be fully excluded that these effects were mediated, in part, by changes of the opposite sign in neurons that were tuned to untrained control orientations. However, some effects of conditioning were quite selective for the trained orientation as compared to different untrained control (i.e. oblique and orthogonal) orientations (see e.g. *Figures 4e* and *5*). While this does not fully confirm that learning effects were only present in neurons tuned to the trained orientation, it indicates that, at least, the trained orientation stood out from the way the broader population of untrained stimuli was processed.

## Sharper spatial tuning within the V1 retinotopic map

Cortical representations, as also found in the auditory cortical tonotopic map or the somatosensory homunculus, have been known to exhibit plasticity (*Bakin and Weinberger, 1996*; *Buonomano and Merzenich, 1998*; *Bao et al., 2001*). An example of map plasticity like observed here, however, has not been found before in visual cortex, where shifts in retinotopic organization were, until now, described as a consequence of selectively depriving cortical input only (*Heinen and Skavenski, 1991*; *Gilbert and Wiesel, 1992*; *Keck et al., 2008*). The kind of refinement of the retinotopic map as shown in our experiments—a sharper retinotopic gradient (*Figure 3e*) in calcium response patterns coupled to a reduced overlap in intrinsic optical responses (*Figure 2e,f,h*) and better location-decoding performance (*Figure 5a*)—is somewhat comparable to emerging discontinuities in the cortical digit representation after separation of webbed fingers (*Mogilner et al., 1993*) or fading discontinuities in the somatosensory map after surgical syndactyly (*Allard et al., 1991*). However, map-plasticity in these studies emerged by self-organizing plasticity, while in our study, appetitive conditioning directed the alterations in the retinotopic map. Not only is the treatment (learning paradigm) more subtle than the interventions in the studies mentioned above, but the observed map-plasticity underscores that also under naturalistic conditions stimulus-reward learning can direct functional changes in cortical maps, expressed in the way the cells in these maps encode sensory information.

The changes in neuronal tuning observed in our study can also be considered in the context of perceptual learning. Training to discriminate small differences in stimulus parameters can result in sharpening of tuning curves in the primary visual cortex (*Schoups et al., 2001*), an effect that has been suggested to help disambiguate fine differences in stimulus orientation. The presently reported learning-induced reduction in overlap of the population response to trained stimuli resembles such a mechanism of tuning curve sharpening, but now pertains to the cortical mapping of visual space. While our study did not directly compare the conditioned group with a sham-trained (randomly rewarded) group, we did observe a correlation between the amount of anticipatory behavior and changes in neuronal population coding (*Figure 5*), but no correlation to the number of trials. This argues that the observed effects are not merely the result of stimulus exposure but are associated with behavioral learning in the conditioning paradigm.

## Mechanisms underlying learning-induced map plasticity in V1

Neuronal learning mechanisms underlying classic forms of map plasticity are generally thought to be Hebbian in nature (*Buonomano and Merzenich, 1998*), functioning through rapid induction of LTP and LTD for spared and deprived inputs respectively or via competition between inputs (*Feldman and Brecht, 2005*). Additionally, local secretion of neurotransmitters like acetylcholine and noradrenaline plays a role (*Bear and Singer, 1986*). In auditory cortex for instance, large shifts in tonotopic representations have been observed after pairing a tone with basal forebrain stimulation (*Bakin and Weinberger, 1996*; *Kilgard and Merzenich, 1998*). In the current study, we did not observe such a large-scale expansion of the conditioned stimulus representation, but rather an increase in selectivity of the cortical map. Recent work in the primary visual cortex of the mouse has established that cholinergic neuromodulation can act to improve neuronal selectivity and perception in a rapid and task specific way (*Pinto et al., 2013*). Moreover, during an appetitive conditioning

eLIFE Research article

Neuroscience

paradigm, cholinergic input from basal forebrain to V1 induced strong local activity patterns leading up to the time of reward delivery (*Liu et al., 2015*), which may drive stimulus-selective plasticity (*Chubykin et al., 2013*). The plasticity in V1's retinotopic map, which we induced here using appetitive conditioning, may therefore have been mediated by local release of acetylcholine. However, the strengthening of feedback connections from higher visual or other cortices (*Pennartz, 1997*; *Makino and Komiyama, 2015*) could provide an alternative or additional mechanism by which learning leads to changes in cortical maps. In addition, the observed reduction in response amplitude to the non-rewarded trained stimulus could indicate a role for an LTD type mechanism reducing excitatory drive of non-rewarded stimulus representation (*Nabavi et al., 2014*). Alternatively, it could reflect strengthening of long-range lateral connections from for example the rewarded retinotopic location, or feed-back from higher visual areas, onto inhibitory neurons in the non-rewarded retinotopic location to increase local inhibitory drive of the non-rewarded stimulus representation (*Makino and Komiyama, 2015*).

While neuromodulation and long-range connectivity are strong candidates to mediate learning in our behavioral paradigm, the conditioning-related improvement in retinotopic selectivity was observed under anesthesia, which makes it likely that these results depend to some degree on changes in feedforward or local recurrent synaptic connectivity. We hypothesize that these cells increase their local connectivity with neurons that are tuned to the same retinotopic location, thereby strengthening their spatial selectivity (see *Figures 2f,h* and *3c–e*) but reducing precise selectivity for stimulus orientation (see *Figure 4f*). From this point of view, it will be interesting to perform an experiment studying the exact receptive field position and structure for each neuron along with its precise local synaptic connectivity. Such an experiment could also reveal whether the receptive field centers of certain subsets of neurons (e.g. cells tuned to the trained orientation) drift away from the border region, or whether the spatial extent of receptive fields selectively shrinks so that it is less likely to cross the trained stimulus border. That receptive fields may have shifted should not be considered a potential confounder, but rather as a part of the underlying mechanism improving discriminability of spatially adjacent cortical representations.

## Learning-related interactions between signal and noise correlations

Several theoretical studies have shown that reduction of noise correlations between cells exhibiting high signal correlations can improve population coding (*Zohary et al., 1994*; *Oram et al., 1998*; *Shadlen and Newsome, 1998*; *Abbott and Dayan, 1999*; *Averbeck et al., 2006*; but see *Montijn et al., 2016*), but experimental evidence in support of these mechanisms being employed during learning is only sparsely available (*Jeanne et al., 2013*). Here, we show a decrease of noise correlations within high signal correlation pairs that was strongest at the V1 location where the trained stimuli bordered (*Figure 6b*). This provides support for the theoretical notion of how interactions between signal- and noise correlations can facilitate coding of stimulus information, and also shows that these effects can arise in the primary visual cortex as a result of associative learning. In addition, groups of cells exhibiting low (negative) signal correlations may improve population coding by increasing noise correlations, even without transmitting information in the structure of the noise correlation itself (*Oram et al., 1998*; *Abbott and Dayan, 1999*; *Averbeck et al., 2006*). Additionally, in imaging regions further away from the border of the trained stimuli, we observed stronger noise correlations in groups of trained orientation tuned cells with relatively low signal correlations. Possibly this effect depends on how heterogeneously a population of neurons is tuned to the behaviorally relevant stimulus and therefore only manifested in the Full imaging regions where the majority neurons might have been synchronously driven by a single (rewarded or non-rewarded) stimulus.

## A less dense, but more efficient cortical representation for reward-associated stimuli

With or without stimulus awareness, stimulus-reward pairing can enhance feature detection in primary visual cortex (*Shuler and Bear, 2006*; *Seitz et al., 2009*; *Frankó et al., 2010*; *Goltstein et al., 2013*). While we observed that visual conditioning resulted in a reduced fraction of neurons being tuned to the trained orientation in the rewarded location (*Figures 3d* and *4d*), those neurons that remained to prefer the conditioned or similar orientations responded with larger ΔF/F amplitudes to the reward associated stimulus (*Figure 4e*). This indicates that population coding for the conditioned

orientation in the rewarded location, contained fewer neurons compared to their untrained orientation tuned counterparts, while maintaining the average population response amplitude, thus potentially rendering their response more efficient by representing the same information using fewer cells (see also *Gdalyahu et al., 2012*).

In summary, this study highlights a selective role for reward-dependent learning in refining the primary cortical retinotopic map. It demonstrates how coordinated changes in neuronal tuning and inter-neuronal correlations can result in a more optimal population code for trained stimuli and improve the local representation of visual space while leaving the overall retinotopic map intact.

## Materials and methods

### Animals

Experiments were conducted with approval of the ethical committee for animal experiments of the University of Amsterdam. Male C57BL/6 mice (for *n*, see *Table 1*; Harlan, The Netherlands) ranging in age from 70 to 116 days at the start of the experiment were housed pair-wise in large cages (40 cm length ×25 cm width ×25 cm height) on a reversed day night cycle (9 AM light-off; 9 PM light-on). Water and food were available *ad libitum*, except for a period of six to eight hours prior to behavioral training, when food was not available. Non head-fixed habituation of the mice to the conditioning environment and the frame for head-restraint spanned four to five days and included handling in the behavioral lab and free access to the reward substance, a semifluid vanilla-flavored dairy product (*Figure 1—figure supplement 1a*).

### Visual conditioning of head-restrained mice

Mice were fitted with a stainless-steel cranial plate that allowed repeated head-restraint (*Figure 1—figure supplement 1a*; *Dombeck et al., 2007*; *Niell and Stryker, 2010*). In brief, under surgical anesthesia (Isoflurane inhalation, induction: 3% in $O_2$; maintenance: 1.5% in $O_2$) and analgesia (Buprenorphine, 0.05 mg $kg^{-1}$ bodyweight, injected subcutaneously 30 min before surgery), three skull screws were placed, two in the frontal skull plates and one lateral-right in the occipital skull plate. The head bar was positioned with the central opening over cortical area V1 and attached with cyanoacrylate glue and black-pigmented dental cement to the skull and screws. The central opening was closed with a transparent silicone elastomer (Quick-sil, World Precision Instruments, Germany) and a cover glass. Mice were allowed to recover for four days, during which non head-fixed habituation was continued. In the next four to five days, animals were head-fixed in the behavioral apparatus while given free access to the vanilla-flavored reward through the feeding tube for 15 to 20 min per day.

Appetitive conditioning with retinotopically selective visual stimuli was done in 8 to 17 sessions. The number of conditioning sessions (8 to 17) before two-photon imaging depended on the order in which animals were used for calcium imaging, which was determined by chance. The number of trials ranged from 8 to 12 trials in the first training session to 20–35 in subsequent training sessions. The amount of rewarded and non-rewarded trials in each training session was nearly equal; on a single day, the number of rewarded and non-rewarded trials never differed by more than three trials and the total number of rewarded and non-rewarded trials differed by only 0.5% (3.1 out of 539.7 trials). The total number of conditioning trials that each mouse was exposed to is reported in *Table 1*. Conditioning sessions lasted 20–30 min, during which the animal was head-restrained facing a 21' computer screen (Dell) at a distance of 20 cm (*Figure 1a,b*). The center of the screen was positioned 45° lateral (Azimuth) at the height of the eye of the mouse (0° elevation). Stimuli were oriented gratings (100% contrast square wave; spatial frequency: 0.05 cycles $degree^{-1}$; temporal frequency: two cycles $second^{-1}$) moving in a single direction, presented either in an area of 30 by 30 retinal degrees above, or below the horizontal meridian (*Figure 1a*, right panels). For each mouse, the rewarded stimulus was randomly assigned to one visual field location and the non-rewarded to the other (see *Table 1*). This assignment was varied across mice, with the rewarded stimulus being as often above the meridian as below it. Stimulus orientation and direction of the rewarded and non-rewarded stimuli were the same and all parameters remained unchanged during the entire behavioral experiment. Each rewarded or non-rewarded trial started with a click, followed by 4 to 6 s of stimulus presentation against a background with equal luminescence. Next followed a pre-reward delay of 1.5 to 2 s.

In the last 0.4 s of the pre-reward delay a servo-controlled arm moved a small plastic tube to a position within licking distance of the mouse in rewarded trials, while the arm stopped just out of range of the mouse in non-rewarded trials (*Figure 1b,d* and *Figure 1—figure supplement 1b–e*). By applying back-pressure on the tube using a syringe pump (RE-S, RAZEL, United Kingdom), an amount of 20–40 µl reward substance was delivered in each rewarded trial. After a reward consumption period of 4 s, the arm moved back and an inter-trial interval of 20 to 30 s was inserted between subsequent trials. The temporal offset between reward delivery and visual stimulation was implemented to prevent consumption-related eye-movements or other motion artifacts during visual stimulus presentation.

Behavioral parameters were monitored using a standard visible light CCD camera fitted with a wide-field low-magnification lens at a frame rate of 25 Hz (*Figure 1*, 'Behavior tracking'). This camera was positioned on the left side of the mouse to record its silhouette and that of the reward arm against the grey background of the screen, as well as to record the rewarded and non-rewarded stimulus presentation (*Figure 1b* and *Figure 1—figure supplement 1b,d*). Behavioral parameters were aligned to stimulus onset and (non-) reward delivery time, which was determined by the arm arriving at the endpoint (*Figure 1b* and *Figure 1—figure supplement 1c,e*). The trajectory and speed of the arm was identical in rewarded and non-rewarded trials up to the point where, in rewarded trials, the arm exceeded the non-reward endpoint. The traversal from the non-reward endpoint to the reward endpoint happened within a single 40 ms image frame (*Figure 1—figure supplement 1f*). The measured time-difference between detection of the arm arriving at the non-reward endpoint and detection at the reward endpoint across all trials was on average 55 ms, implying a movement speed between 0.5 and 1.0 m s$^{-1}$. We added an extra margin of 100 ms to the 55 ms difference in end point arrival time to buffer possible variations in movement speed. As a result, we excluded behavioral data in the period from 160 ms before the reward or non-reward endpoint in the comparison of learning effects.

Nasal movements (putative sniffs) and oral movements (putative licks) were quantified by applying a threshold to each video frame at the mean intensity, resulting in a binary silhouette of the mouse, and counting the number of pixels that changed intensity compared to the previous video frame in the nasal or oral region (*Figure 1c* and *Figure 1—figure supplement 2a*, red box). Licks are rapid, sparsely occurring events; therefore a threshold was applied to the number of pixels changed in each frame, which resulted in 'all or none' detection of oral movement. Sniffs are slower and more continuous events that are better reflected by a continuous measure. We therefore used the mean number of pixels that changed intensity as a consequence of nasal-movement directly as a measure for sniffing.

## Eye-tracking

Position and diameter of the pupil was monitored using a near infrared CCD camera (Jai, United States) through a high magnification long working distance lens (Thorlabs, Germany) with a frame rate of 25 Hz (see *Figure 1* for camera position at 'Eye tracking'). Eye-tracking was done offline using a custom written algorithm (*Zoccolan et al., 2010*). For each video frame, location of the center of the pupil was estimated by applying a radial symmetry transform (*Loy and Zelinsky, 2003*). The edge of the pupil was detected from this point of origin into all directions with 10° angular resolution. Outlier edge detection points (beyond mean ±3 SD) were removed and the remaining data points were fitted with a circle (*Figure 1—figure supplement 3a*). This procedure provides estimates for the X and Y position and diameter of the pupil in pixels. Using the estimated average diameter of the mouse eye (3.3 mm; *Jeon et al., 1998*) and the field of view of our camera, a 1-pixel displacement/enlargement in pupil-tracking data was estimated to correspond to approximately 2–2.5 retinal degrees of visual field.

## Transcranial optical imaging of intrinsic signals

Intrinsic optical imaging (*Bonhoeffer et al., 1996*) was performed using a commercially available setup (Optical Imaging Ltd, Israel). The mouse was head-restrained while being under anesthesia (isoflurane inhalation, induction: 3% in $O_2$; maintenance: 1–1.5% in $O_2$). This procedure kept the animal immobile, whereas the eye-blink reflex was just noticeable. The skull was illuminated through the cover glass with 630 nm (bandpass filtered) light. Images were acquired at an effective sampling

rate of 2 Hz. A screen was placed at the exact same position as in the behavioral setup (see *Figure 1a*). Five stimuli (the rewarded stimulus, the non-rewarded stimulus, stimuli at the same location but with orthogonal orientations, and an isoluminant grey screen, the null stimulus; *Figure 2a*, left panels) were presented in 20 trials each. During the entire imaging experiment, the screen displayed the isoluminant null stimulus, except when a grating stimulus was presented. Each trial started with the recording of a four-second baseline period, followed by four seconds of continuous stimulus presentation and seven seconds of post-stimulus response time. The intertrial interval lasted an additional five seconds.

## Analysis of intrinsic optical signals

Intrinsic signal images were referenced per trial against the mean of the first 8 (baseline) frames, resulting in a percentage signal increase per pixel. A region in the primary visual cortex, large enough to contain the responses to both rewarded and non-rewarded stimuli, was selected manually and within-trial signal drift was corrected for by subtracting the value of each pixel in this region by the average intensity of a non-responsive image-region a few millimeters anterior. For each recording, a single mean and standard deviation was calculated per pixel across all stimuli and repetitions and each pixel was subsequently z-scored by subtracting the mean and dividing over the standard deviation.

Within the region of V1, the location of maximum activation for the rewarded and non-rewarded area was automatically detected by locating the largest activation in smoothed average response images for visual stimulation by the orthogonal stimuli in the rewarded area and non-rewarded area. Using this (untrained) template response map, we calculated the angle that aligned an imaginary line, connecting peaks of maximal visual activity evoked by stimulation in the different visual fields, parallel to the vertical axis of the template image (*Figure 2a*, red windows). In most mice this rotation was around 25° clockwise. All further data from each recording was automatically rotated to the calculated rotation angle. Next, cutouts of this 2-dimensional rotated image, from −5 to +5 pixels centered on the vertical transection of the average peak responses in the (untrained) template, were averaged to a 1-dimensional spatial response profile for each time point separately that represented response strength as a function of cortical distance aligned to the Rewarded/Non-rewarded axis (*Figure 2b*, left panels, cutout before averaging the x-dimension). Response profiles from different time points (imaging frames) of the same trial were concatenated to cortical distance ×time matrices (*Figure 2b*, right panels). On a total of 10 mice, one mouse was removed from these analyses because the peak location of the intrinsic response could not be (automatically) detected in both the rewarded and the non-rewarded location due to blood vessel response artifacts in the imaging data.

Amplitude time courses (*Figure 2—figure supplement 1b*) of the intrinsic optical signal response were calculated by averaging values of pixels in the spatial dimension over a range of −5 to +5 pixels around the maximum pixel in the cortical distance ×time matrices. Amplitude-cortical space profiles (*Figure 2d*) were constructed similarly, by averaging intensity values in the period where the response peaked (i.e. 3 to 7 s after stimulus onset) over the time dimension. Overlap of cortical space profiles was calculated by, for each spatial point on the profile, subtracting the sum of the individual profile data points from their maximum, resulting in the part of the response that was shared (see *Figure 2c* and *Figure 2f*). ∆Overlap (*Figure 2h*) indicated the difference between overlap calculated for trained and for orthogonal stimulus orientations presented at the same location. Spatial selectivity was calculated similarly by, for each spatial point on the profile, dividing the difference of the responses to the rewarded ($R_{rewarded}$) and non-rewarded ($R_{non-rewarded}$) locations by their sum $(R_{rewarded} - R_{non-rewarded}) / (R_{rewarded} + R_{non-rewarded})$.

## Calcium imaging

Under surgical anesthesia (Isoflurane inhalation, induction: 3% in $O_2$; maintenance: 1.5% in $O_2$) and analgesia (Buprenorphine, 0.05 mg kg$^{-1}$ bodyweight, injected subcutaneously 30 min before surgery), a craniotomy was performed over the area of V1 that was identified to respond to the trained stimuli using intrinsic optical imaging. In these regions, cells were loaded with the fluorescent calcium indicator Oregon Green BAPTA1-AM (OGB) and Sulforhodamine 101 (SR101; for staining astrocytes/glial cells) using a protocol for multi-cell bolus loading (*Stosiek et al., 2003*; *Nimmerjahn et al., 2004*; *Goltstein et al., 2013*; *Goltstein et al., 2015*). After surgical procedures

were completed, the Isoflurane concentration was carefully lowered and maintained at 0.8% for the entire course of the experiment (such that the mouse remained under anesthesia).

Two-photon laser scanning microscopy was performed with a Leica SP5 resonant scanner and a Spectra-Physics Mai Tai High Performance Mode Locked Ti:Sapphire laser (wavelength 810 nm). Fluorescence was collected in non-descanned photo-multiplier tubes, filtered at 525 nm (maximum range 500–550 nm) for OGB and 585 nm (maximum range 565–605 nm) for SR101. Frame-averaged images (eight frames, 512 × 512 pixels) from a square region of approximately 330 × 330 μm were acquired at a scan speed of approximately 2.8 Hz. Visual stimulation was done with moving gratings identical to those used for behavioral conditioning, but now 8 directions of movement (four orientations) were shown in either visual field (10 repetitions with a duration of 8 scan frames per stimulus).

## Analysis of calcium imaging data

Images acquired with two-photon microscopy were realigned to compensate for small movement artifacts in the x–y plane using an algorithm that relies on the cross-correlation between the two-dimensional Fourier transforms of the to-be-aligned image and a reference image (*Guizar-Sicairos et al., 2008*). Neurons were manually identified and outlined and the mean fluorescence was calculated for each image frame. The fluorescence time series of each neuron was corrected for possible contamination by non cell-specific fluorescence signals that originate from the neuropil using previously described methods (*Kerlin et al., 2010*; *Goltstein et al., 2015*). Visual responses were quantified per trial as $\Delta F/F$, or in more detail: $(F–F_0)/F_0$. F was determined from the average fluorescence in the 8-frame ($\sim$2.8 s) period during visual stimulation. $F_0$ was calculated from the average fluorescence in the 8-frame period preceding visual stimulation, when a gray screen with equal luminance was presented.

For each field of view, an imaging location index was calculated from the mean response of all neurons to the orthogonal (untrained) orientations (*Equation 1*). This index approaches the value of $-1$ for fields of view that have primarily cells tuned to the non-rewarded location and +1 when most cells are tuned to the rewarded location. For fields of view where the fraction of cells responding to the rewarded and non-rewarded location was similar the index has a value around 0:

$$Imaging\ location\ index = \frac{\left(\frac{1}{N}\sum_{1}^{N}R_{Orth(RewLoc)_n}\right) - \left(\frac{1}{N}\sum_{1}^{N}R_{Orth(nRwLoc)_n}\right)}{\left(\frac{1}{N}\sum_{1}^{N}R_{Orth(RewLoc)_n}\right) + \left(\frac{1}{N}\sum_{1}^{N}R_{Orth(nRwLoc)_n}\right)} \tag{1}$$

Where N is the total number of neurons in a recording session and $R_{Orth(RewLoc)n}$ is the mean response of the $n^{th}$ neuron to the orientation orthogonal to the trained orientation, presented in the reward-predictive visual field.

Based on the imaging location index, the dataset was divided in four groups along the rewarded/non-rewarded axis (*Figure 3b*). Two groups consisted of recordings that were made in regions that were mostly driven by one retinotopic location; the 'Full rewarded' and 'Full non-rewarded' groups (Imaging location index $<-0.66$ for Full-nRw and $>0.66$ for Full-Rew). The two other groups consisted of recordings obtained from the region of cortex where the rewarded and non-rewarded retinotopic representations bordered; the 'Border rewarded' and 'Border non-rewarded' groups (Imaging location index $>-0.66$ but $<0$ for Border-nRw and $>0$ but$<0.66$ for Border-Rew). The cut-off of 0.66 was chosen such that only cortical regions that were strongly driven by one stimulus location compared to the other stimulus location ($\sim$5 times stronger) would be classified as Full rewarded or Full non-rewarded.

The number of recordings in each location group ranged from 8 to 14 (*Figure 3—figure supplement 1a*) and the average within-group imaging-location index showed a linear increase across location groups along the rewarded/non-rewarded axis (*Figure 3—figure supplement 1b*). The latter indicates that each location-group represented a part of the visual space that was approximately equally distant from the neighboring location groups in cortical space and thus validates the approach. An important aspect here is that, as in the analysis of intrinsic imaging data, the imaging location index was calculated using responses to untrained stimuli, which facilitates the comparison between results obtained using these two imaging techniques.

Neurons were classified as being orientation-tuned or non-tuned by testing for a difference in their responses across eight directions in the upper or lower visual field separately using a 1-way Anova against p<0.05. Neurons that displayed significant orientation tuning, but for which the ΔF/F response to the preferred direction did not exceed 2%, were considered to be false positives, or to have a low signal to noise ratio and were therefore discarded from the analysis. Tuning curves for movement direction were constructed by fitting the mean responses across all directions in each area separately by a two-peaked Gaussian function (*Li et al., 2008*). Tuning curve bandwidth was calculated as the half-width of the fitted tuning curve at $1/\sqrt{2}$ height (*Ringach et al., 2002*). Orientation selectivity (OSI) was calculated by dividing the difference between response to the preferred orientation and the orthogonal orientation by the sum of these: ($R_{pref-ori}$ - $R_{ortho-ori}$)/($R_{pref-ori}$ + $R_{ortho-ori}$) (*Niell and Stryker, 2008*).

Population sparseness $a^P$ (*Equation 2*) was quantified as described in *Rolls and Treves, 2011*. The vector $y$ contains mean ΔF/F responses of all $N$ neurons to a single stimulus and $y_n$ is the response of a single neuron $n$ to a single stimulus. Higher values of $a^P$ indicate that many neurons in the population were responsive to the stimulus. Lower values of $a^P$ indicate a sparser population response, with only few strongly responding neurons resulting in a more skewed distribution of ΔF/F amplitudes:

$$a^p = \frac{\left(\sum_{n=1}^{N} y_n/N\right)^2}{\left(\sum_{n=1}^{N} y_n^2\right)/N} \qquad (2)$$

Population decoding of the stimulated visual field (rewarded or non-rewarded location), was done using an algorithm based on Bayes theorem (*Equation 3*; *Oram et al., 1998*; *Dayan and Abbott, 2001*). Single trial responses were decoded using a 'leave one trial-block out' (cross-validated) principle. One trial from each stimulus category (rewarded or non-rewarded location) was removed from the dataset and each of the removed trials was decoded separately. The probability for a single-trial calcium response $r$ of an individual neuron to be observed in a certain stimulus condition $s$, $p(r|s)$, was calculated from a Gaussian probability distribution, estimated by the mean and SD of all remaining responses of the decoded neuron in that stimulus condition. When multiple neurons were used for decoding, probabilities of the individual neurons' responses were multiplied. The prior, $p(s)$, was identical across stimuli and the overall probability of observing the decoded response, $p(r)$, was estimated from all remaining responses (*Equation 3*; *Oram et al., 1998*). This procedure gave two probabilities per decoded trial; one for the stimulation delivered at the rewarded location and one for the non-rewarded location. The highest probability was chosen as decoder output. Performance for decoding single neuron or population (multiple simultaneously recorded neurons) calcium responses was quantified by comparing the decoder output with actual stimulus position and resulted in a fraction correct (the theoretical chance level of 50% correct was experimentally confirmed).

$$p(s|r) = \frac{p(r|s)p(s)}{p(r)} \qquad (3)$$

Signal correlations between pairs of cells were quantified by calculating the mean Pearson correlation coefficient of the average responses of those cells to all 16 stimulus types (eight directions × 2 locations). Noise correlations were quantified by subtracting the average response to the preferred orientation from the individual trial responses to that orientation (because of 8 repeats per stimulus for the rewarded and non-rewarded location this resulted in a vector of 16 mean-subtracted single trial responses), and subsequently calculating the Pearson correlation coefficient between the mean-subtracted single trial responses of the pair of cells, but only for the preferred stimulus in the rewarded and non-rewarded location (*Hofer et al., 2011*; *Jeanne et al., 2013*).

## Statistics

All data are presented and tested as mean ± SEM across mice unless otherwise noted. Most parameters that were reported in this manuscript were likely not-normally distributed and were therefore tested with a Mann-Whitney U test or a Wilcoxon matched-pairs signed-rank test (WMPSR). When multiple groups were involved, we applied a Kruskal-Wallis test, with posthoc Mann-Whitney U tests

or WMPSR tests; or an Anova with posthoc t-tests if the data followed a normal distribution (*Figure 3d,e*). When iterating comparisons between multiple groups without first using a test for multiple groups (Anova/Kruskal-Wallis test), we corrected the P-value using Bonferroni's method.

Signal- and noise correlations (*Figure 6*) were calculated per imaging field of view, on all simultaneously recorded pairs of neurons. Next, 95% and 99% confidence intervals were calculated using bootstrap resampling on the resulting dataset with signal- or noise correlations from all fields of view (resampling with replacement; Sample size of 250 pairs; 10000 resamples; *Efron, 1979*).

## Acknowledgements

The authors thank Laura van Mourik-Donga for technical assistance and Lior Cohen, Jorrit Montijn, Mark Hübener and Tobias Rose for helpful comments on the manuscript. This work was supported by the NWO Program for Excellence, Brain and Cognition, 433-09-208, NWO VICI Grant 918.46.609 and the European Union's Horizon 2020 Research and Innovation Program under Grant Agreement No. 785907 (Human Brain Project SGA2).

## Additional information

### Funding

| Funder | Grant reference number | Author |
| --- | --- | --- |
| Nederlandse Organisatie voor Wetenschappelijk Onderzoek | 433-09-208 | Cyriel MA Pennartz |
| Horizon 2020 Framework Programme | Grant Agreement No. 785907 (Human Brain Project SGA2) | Cyriel MA Pennartz |
| Nederlandse Organisatie voor Wetenschappelijk Onderzoek | 918.46.609 | Cyriel MA Pennartz |

The funders had no role in study design, data collection and interpretation, or the decision to submit the work for publication.

### Author contributions

Pieter M Goltstein, Conceptualization, Data curation, Software, Formal analysis, Investigation, Visualization, Methodology, Writing—original draft, Writing—review and editing; Guido T Meijer, Data curation, Software, Formal analysis, Investigation, Methodology, Writing—review and editing; Cyriel MA Pennartz, Conceptualization, Supervision, Funding acquisition, Writing—original draft, Project administration, Writing—review and editing

### Author ORCIDs

Pieter M Goltstein (ID) https://orcid.org/0000-0001-8895-0841
Cyriel MA Pennartz (ID) http://orcid.org/0000-0001-8328-1175

### Ethics

Animal experimentation: Experiments were conducted with approval of the ethical committee for animal experiments of the University of Amsterdam (Protocol number DED162).

### Decision letter and Author response

Decision letter https://doi.org/10.7554/eLife.37683.019
Author response https://doi.org/10.7554/eLife.37683.020

## Additional files

### Supplementary files

• Transparent reporting form

DOI: https://doi.org/10.7554/eLife.37683.015

### Data availability

The full dataset of this study is available on gin.g-node, url: https://web.gin.g-node.org/pgoltstein/goltstein_meijer_pennartz_2018

The following dataset was generated:

| Author(s) | Year | Dataset title | Dataset URL | Database, license, and accessibility information |
|---|---|---|---|---|
| Pieter M Goltstein, Guido T Meijer, Cyriel MA Pennartz | 2010 | Behavioral-, intrinsic imaging-, calcium imaging- and eye-tracking-data | https://web.gin.g-node.org/pgoltstein/goltstein_meijer_pennartz_2018 | Publicly available at the Gin: Modern Research Data Management for Neuroscience website. |

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
