## [Decision Letter]

Thank you for submitting your article "Conditioning sharpens the spatial representation of rewarded stimuli in mouse primary visual cortex" for consideration by *eLife*. Your article has been reviewed by three peer reviewers, including Daeyeol Lee as the Reviewing Editor and Reviewer #1, and the evaluation has been overseen by David Van Essen as the Senior Editor.

The reviewers have discussed the reviews with one another and the Reviewing Editor has drafted this decision to help you prepare a revised submission.

Summary:

Goldstein et al., examined how the neural representation of visual stimuli in the mouse primary visual cortex (V1) might be altered during classical conditioning, using both intrinsic optimal imaging and calcium imaging methods. For the intrinsic optical imaging, they found that the overlap of the activity induced by visual stimulus in the same orientation used during conditioning across the cortical surface was reduced compared to the overlap for the stimulus in the orthogonal orientation. This might suggest that the conditioning improved the quality of location information specifically for the experienced orientation. They obtained somewhat similar results from the calcium imaging of individual V1 neurons. First, they found that the overlap was smaller in all the locations other than the "full-non-rewarded" location (however, see below). Second, they found that the fraction of neurons tuned to the tested orientation was reduced, but their response strength increased, although this effect looked relatively small (Figure 4B). Finally, they showed that the encoding of spatial information was enhanced more for the tested orientation than for untested orientations.

Overall, this study addresses an important question regarding how the cortical representation of sensory stimuli might be affected by their behavioral relevance. The experiments and analyses were carried out rigorously, and multiple analytical approaches were applied thoroughly. Nevertheless, there are some apparent inconsistencies in the main findings, which should be addressed or discussed better by the authors.

Essential revisions:

1) The most significant weakness of this manuscript is its heavy reliance on the measure related to the changes in the overlap in the neural activity induced by different visual stimuli across the cortical surface.

For the intrinsic imaging data, the authors focus on the changes in the overlap in the neural activity across the cortical surface. Thanks to the data presented in Figure 2*—*figure supplement 1, however, it can be seen that in reality the data are a bit more complicated. In particular, the overlap was reduced mainly because the activity for the stimulus in the non-rewarded location was reduced in the cortical area corresponding to the rewarded location. By contrast, the activity related to the conditioned stimulus in the rewarded location did not change. If one assumes that learning is driven by stimulus-reward association during conditioning, this result is counter-intuitive. The authors should discuss this finding more directly, rather than focusing on the change in the overlap instead.

The results from the intrinsic and calcium imaging experiments are also only superficially similar. The reduction in the overlap for tested orientation is driven mostly change in the "full" non-rewarded location (Figure 3C and 3D). Again, this is counter-intuitive and problematic, since it would be expected the results for un-rewarded stimulus would provide the baseline. In addition, the fact the significant change is now in the cortical area corresponding to the non-rewarded location is opposite to the results from the intrinsic optical imaging, where the change was seen in the cortical area corresponding to the rewarded location. Therefore, contrary to the author's assertion, the results from these two imaging modalities are not entirely consistent. This should be discussed more directly.

IOS ΔOverlap is a key measure put forth by the authors in advancing the narrative that conditioning results in a greater separation between the nR and R location stimuli-evoked responses in comparison to that of the non-conditioned (orthogonal) stimuli. They imply with this measure that post conditioning, NR and R stimuli decrease in their overlap; but, as the measure is a difference of differences, I think it can be misleading to refer to the effect observed as such. For instance, let us suppose that conditioning was to elevate (across its spatial extent) the IOS response to the R stimulus and decrease the response to the nR. The resulting "mountain" of overlap for the conditioned case would be shifted toward the nR location. By subtracting it from the mountain of overlap for the ortho case (which is used as a proxy for what might be expected were the conditioned orientation to be pseudo-conditioned), a profile of ΔOverlap like that shown in 2D bottom would result. So, the observed profile can arise not from increasing the spatial segregation of the nR and R stimuli in the conditioned case, but rather from an increase in the R stimulus evoked response across its spatial extent and a decrease in the NR response across its spatial extent. So, what then is the reported cause underlying the ΔOverlap? The authors use of the spatial selectivity index indicates that a feature of conditioning was that the amplitude of the nR was decreased. So, while I find the ΔOverlap a somewhat obtuse manner of assessing the data, and how it is presented and discussed to be somewhat confusing, these findings do support the conclusion that "conditioning using rewarded and non-rewarded stimuli occupying neighboring regions of visual space selectively affect the retinotopic organization for the conditioned stimuli compared to the control stimuli". It appears from these data that the main driver of the effect was to diminish the evoked response to the nR stim across its spatial extent in the conditioned case (and from Figure 2—figure supplement 2C bottom left, from elevating the response evoked by the R stim at the NR location).

2) The analysis on the fraction of tuned neurons might be flawed (Figure 4). In particular, the left panel of Figure 4A shows that the fraction of orientation neurons was higher in the cortical region for the rewarded location, and the results in the right panel of Figure 4B are normalized, presumably according to the overall proportion for each cortical area (since their averages look comparable for both regions). This implies that the main change might be for the untested orientations in non-rewarded region. Again, this is counter-intuitive and would be inconsistent with the author's conclusion that the representation of the rewarded stimulus became sparser.

3) The analysis presented in the manuscript does not convincingly demonstrate that the representation of conditioned stimulus became sparser and stronger at the same time. In order to show this, it would be better to examine the entire distribution of tuning strengths across the entire population, rather than calculating the sparsity and strength using an arbitrary criterion.

---

## [Author Response]

[…] Essential revisions:1) The most significant weakness of this manuscript is its heavy reliance on the measure related to the changes in the overlap in the neural activity induced by different visual stimuli across the cortical surface.For the intrinsic imaging data, the authors focus on the changes in the overlap in the neural activity across the cortical surface. Thanks to the data presented in Figure 2—figure supplement 1, however, it can be seen that in reality the data are a bit more complicated. […] The authors should discuss this finding more directly, rather than focusing on the change in the overlap instead.The results from the intrinsic and calcium imaging experiments are also only superficially similar. The reduction in the overlap for tested orientation is driven mostly change in the "full" non-rewarded location (Figure 3C and 3D). Again, this is counter-intuitive and problematic, since it would be expected the results for un-rewarded stimulus would provide the baseline. In addition, the fact the significant change is now in the cortical area corresponding to the non-rewarded location is opposite to the results from the intrinsic optical imaging, where the change was seen in the cortical area corresponding to the rewarded location. Therefore, contrary to the author's assertion, the results from these two imaging modalities are not entirely consistent. This should be discussed more directly.IOS ΔOverlap is a key measure put forth by the authors in advancing the narrative that conditioning results in a greater separation between the nR and R location stimuli-evoked responses in comparison to that of the non-conditioned (orthogonal) stimuli. They imply with this measure that post conditioning, NR and R stimuli decrease in their overlap; but, as the measure is a difference of differences, I think it can be misleading to refer to the effect observed as such. […] It appears from these data that the main driver of the effect was to diminish the evoked response to the nR stim across its spatial extent in the conditioned case (and from Figure 2—figure supplement 2C bottom left, from elevating the response evoked by the R stim at the NR location).

From this comment we understand that the reviewers have concerns regarding (1) the interpretation of the intrinsic imaging data (in particular the measure of Overlap) which we address first, and (2) with the link between the intrinsic imaging data and the calcium imaging data as presented in Figure 3, which we will discuss further below.

Regarding the intrinsic imaging data:

The reviewers suggested that, conceptually, the change in response amplitude may be the most basic parameter describing the learning-related changes in intrinsic signal responses, as opposed to the derived parameter of Overlap. We agree with the reviewers that the reduction in response amplitude to the non-rewarded trained stimulus is an important aspect of the change after conditioning and have therefore adjusted the order in which we present the data. We now clearly and directly indicate in the first paragraph of the subsection “Mesoscopic shifts in cortical representations of conditioned stimuli” that we observe a reduction in response amplitude to the trained non-rewarded stimulus (as compared to the orthogonal). In order to facilitate interpretation of the effect by the reader, we have added the visualization of the average real data response profiles (originally presented in Figure 2—figure supplement 2C) in the main Figure 2D and we have added a Figure panel (2G) showing the average response amplitude across the different imaging time points (controlled for the amplitude to the orthogonal control stimuli).

In the next paragraph of the manuscript (subsection “Mesoscopic shifts in cortical representations of conditioned stimuli”), we describe how the change in response amplitude can lead to a reduction in overlap of the cortical representations and again facilitate this by showing now the overlap-profiles of all time points (originally presented in Figure 2—figure supplement 2D) in the main Figure (2F). In addition, Figure 2H shows the average difference in overlap between conditioned and control stimuli across the consecutive imaging time points. We believe that by presenting the data more extensively and in this particular order, the reader should be able to better judge how the intrinsic signal response changes as result of conditioning and better understand how overlap is computed from the ‘raw’ response amplitude.

Furthermore, as the reviewers mentioned, the measure of overlap is indeed sensitive to the ‘raw’ response amplitude. We still show the measure of Overlap because we think that it captures best how the two cortico-topically adjacent representations segregate. But by now displaying the curves of Figure 2D and 2F directly below each other in the main figure, the reader can see better how the overlap depends on the amplitude. Moreover, we have (1) acknowledged the potential confounder of response amplitude directly in the manuscript in subsection and (2) presented the results obtained using the ratiometric spatial selectivity index now also visually in the main Figure 2I.

In order to more explicitly address the question of whether the representation of the Rewarded and Non-rewarded stimulus drifted apart in cortical space (i.e. by a translation on the cortical plane, not an amplitude modulation), we have added an analysis of the distance between the peaks of the conditioned stimulus representations (subsection “Spatial organization of neuronal population activity for conditioned stimuli”). While after conditioning they appeared to be slightly further apart compared to the control orientations, this was not significantly different from chance.

In addition, we would like to point out that all three measures (response amplitude to the non-rewarded conditioned stimulus, overlap and spatial selectivity index) are strongly correlated with each other and therefore most likely reflect closely related mechanisms operating on both the rewarded and non-rewarded side of the conditioned stimulus representation (subsection “Spatial organization of neuronal population activity for conditioned stimuli”).

Thus, we observed closely related changes in overlap, spatial selectivity index and the response amplitude to the non-rewarded trained stimulus. That the response amplitude to the non-rewarded stimulus was reduced after conditioning is our opinion not necessarily counter-intuitive. A reduction in response amplitude might occur if an LTD type mechanism engages during conditioning at times when the non-rewarded stimulus presentation is not followed by reward, thus reducing the overall excitatory drive onto the non-rewarded stimulus representation (see e.g. Navabi et al., 2014). Alternatively, it could reflect a strengthening of feed-back connections from e.g. higher visual areas onto local inhibitory neurons increasing inhibitory drive of the non-rewarded stimulus representation (Makino and Komiyama 2015). These considerations have been added to the Discussion section.

Regarding the link between intrinsic imaging and calcium imaging data:

The reviewers stated that the results obtained using calcium imaging are not consistent with those obtained using intrinsic imaging. Specifically, the reviewers note that the change in Ca-Overlap is on the Full Non-rewarded side. We address this issue by providing better explanation and discussion of the similarities as well as the differences between the two methods:

1) Figure 3C shows the reduction in Ca-Overlap, which is greatest for the Border Non-rewarded and Border Rewarded locations but is also present for the Full Rewarded region. In the intrinsic imaging data, the reduction is indeed more strongly biased to the Rewarded side. In order to prevent further confusion on these results, we have added an extra sentence stating this similarity as well as this difference in the results (subsection “Spatial organization of neuronal population activity for conditioned stimuli”) and added a paragraph on the comparison of intrinsic imaging data in the Discussion section. We would like to note here that both the measures of Overlap and of Spatial selectivity correlated positively between methods (calcium imaging and intrinsic imaging; see subsection “Spatial organization of neuronal population activity for conditioned stimuli”).

2) Figure 3D shows the fraction of tuned neurons, which is a measure of how many neurons are significantly tuned to the orientation of the grating stimuli. This measure is in our opinion not directly comparable with intrinsic imaging data, as it does not take into account the actual response amplitude of the neurons. Therefore, this does not argue in favor, nor against, similarity with the intrinsic imaging data. We adjusted the text on this point in subsection “Orientation-tuned neurons respond more sparsely and strongly to rewarded conditioned stimuli”).

3) Finally, although the calcium imaging response amplitude to the trained non-rewarded stimulus appeared lower compared to the response amplitude of the control orientation in the non-rewarded stimulus location (similar effect as in the intrinsic imaging data, see Figure 3E and inset for comparison), this difference was not significantly different and therefore not entirely consistent. In order to be fully transparent, we specifically point this out in the Results section. We think that this difference between calcium imaging and intrinsic imaging data might, in part, be due to the different nature of the two methods (with intrinsic imaging measuring a mixed signal across layers, reflecting metabolism and oxygen consumption, depending on activity of somata, dendrites, synapses, glia and axons versus calcium activity faithfully reflecting layer 2/3 somatic spiking activity). We have added this point to the Discussion section.

We would like to add that the response amplitudes observed using calcium imaging and intrinsic imaging show differences in the same direction (comparison of main panel with inset in Figure 3E), and that the reduction in response amplitude to the Non-rewarded stimulus (compared to control) is positively correlated in the (few) mice where we have data using both techniques (subsection “Orientation-tuned neurons respond more sparsely and strongly to rewarded conditioned stimuli”).

In conclusion, we acknowledge that the data obtained using intrinsic imaging and calcium imaging are not an exact mirror image of each other, but also argue that the results are comparable to a reasonable degree, especially considering the quite different nature of the two methods. We believe that the data obtained using each method provide evidence suggesting (1) that the spatial selectivity of cortical neuronal activity patterns improves compared to the control stimuli, and (2) that the changes following conditioning are expressed in only a select group of neurons defined by the neurons’ preferred orientation and spatial location on the cortical retinotopic map (this is also stated in subsection “Orientation-tuned neurons respond more sparsely and strongly to rewarded conditioned stimuli”).

2) The analysis on the fraction of tuned neurons might be flawed (Figure 4). In particular, the left panel of Figure 4A shows that the fraction of orientation neurons was higher in the cortical region for the rewarded location, and the results in the right panel of Figure 4B are normalized, presumably according to the overall proportion for each cortical area (since their averages look comparable for both regions). This implies that the main change might be for the untested orientations in non-rewarded region. Again, this is counter-intuitive and would be inconsistent with the author's conclusion that the representation of the rewarded stimulus became sparser.

Figure 4A, left panel, indeed shows that the fraction of neurons that responded to the Rewarded region (17.8%) was slightly (but non-significantly, Wilcoxon Matched-pairs signed-rank test, p=0.69) larger than the fraction of neurons that responded to the non-rewarded region (14.5%). However, the normalization of the fraction of responsive neurons per orientation and rewarded/non-rewarded region (for original Figure 4A) was done using the summed total of all responsive neurons across both rewarded and non-rewarded regions, and all preferred orientations per mouse. The normalization was done such that if the preferred orientations of a mouse would have a flat distribution, each bin would have the value of 1.0 (irrespective of the absolute fraction of responsive cells per mouse, and irrespective of the bin width). We realize that the figure legend described the normalization procedure rather briefly, and wrongly suggested that we normalized data by the mean of the individual orientation bins. The specific reason to perform the normalization per mouse was to ensure that the analysis of the fraction of tuned neurons per orientation/region was not biased by variation in the absolute number of responsive neurons across individual mice and variation in recording regions across mice. Therefore, we would like to emphasize that the actual analysis was not flawed (only the description in the figure legend was possibly unclear). We have added a statement on the overall difference in numbers of responsive neurons in subsection “Improved neuronal population coding for conditioned stimulus location”, expanded Figure 4 to better reflect the absolute and relative fractions of tuned neurons, added explanation to the legend of Figure 4, and added a statement explaining how the normalization was done in subsection “Improved neuronal population coding for conditioned stimulus location”.

As the reviewer indicates, it remains possible that conditioning affected the neuronal representation of the non-trained orientation. The most straightforward way to address this question is to compare timepoints before and after conditioning, but the calcium imaging data have no such control as they were obtained in an acute experiment. The intrinsic imaging experiment was repeated across timepoints, but the response amplitude of the intrinsic signal hardly compares to the fraction of responsive neurons obtained using calcium imaging. However, some effects of conditioning were very selective for cells exactly tuned to the conditioned orientation (e.g. the increased response amplitude to rewarded stimulus presentation in Figure 4E and the improved decoding performance in figure 5A). This argues that (at least) in these cases the rewarded conditioned stimulus representation stood out from the oblique and orthogonal control orientations, increasing the likelihood that changes were expressed in the rewarded stimulus representation selectively, rather than in the general population of neurons excluding the rewarded representation. We have addressed this potential caveat in a paragraph in the Discussion section.

3) The analysis presented in the manuscript does not convincingly demonstrate that the representation of conditioned stimulus became sparser and stronger at the same time. In order to show this, it would be better to examine the entire distribution of tuning strengths across the entire population, rather than calculating the sparsity and strength using an arbitrary criterion.

As the reviewer requested, we have quantified the sparseness of the population tuning curve responses across all cells using the measure described in Rolls and Treves (2011). Although the overall population tuning curve response was slightly sparser for orientations that were more similar to the trained orientation presented in the rewarded region, this difference was not significant. We have added this result to the manuscript in subsection “Effects of conditioning on signal and noise correlations” and as a new panel in Figure 4G.

In order to better describe these data, we have adapted the manuscript to remove the conclusion that the population response was “sparser but stronger”. Instead the manuscript now reports that there were fewer neurons significantly responding to the rewarded orientation in the rewarded location (as is most significantly evident from Figure 3D, but from Figure 4D as well) and that the neurons that were still significantly responding, did so with a larger amplitude (as shown in Figure 4E). This has been updated in the Results section and the Discussion section.